# Flexomagnetism and vertically graded Néel temperature of antiferromagnetic $Cr_2O_3$ thin films

Pavlo Makushko[1,8], Tobias Kosub[1,8], Oleksandr V. Pylypovskyi [1,2] ✉, Natascha Hedrich[3], Jiang Li[1], Alexej Pashkin [1], Stanislav Avdoshenko[4], René Hübner [1], Fabian Ganss[1], Daniel Wolf [5], Axel Lubk [5,6], Maciej Oskar Liedke [7], Maik Butterling [7], Andreas Wagner [7], Kai Wagner[3], Brendan J. Shields[3], Paul Lehmann[3], Igor Veremchuk[1], Jürgen Fassbender [1], Patrick Maletinsky [3] & Denys Makarov [1] ✉

Antiferromagnetic insulators are a prospective materials platform for magnonics, spin superfluidity, THz spintronics, and non-volatile data storage. A magnetomechanical coupling in antiferromagnets offers vast advantages in the control and manipulation of the primary order parameter yet remains largely unexplored. Here, we discover a new member in the family of flexoeffects in thin films of $Cr_2O_3$. We demonstrate that a gradient of mechanical strain can impact the magnetic phase transition resulting in the distribution of the Néel temperature along the thickness of a 50-nm-thick film. The inhomogeneous reduction of the antiferromagnetic order parameter induces a flexomagnetic coefficient of about $15\,\mu_B\,nm^{-2}$. The antiferromagnetic ordering in the inhomogeneously strained films can persist up to 100 °C, rendering $Cr_2O_3$ relevant for industrial electronics applications. Strain gradient in $Cr_2O_3$ thin films enables fundamental research on magnetomechanics and thermodynamics of antiferromagnetic solitons, spin waves and artificial spin ice systems in magnetic materials with continuously graded parameters.

A standard micromagnetic approach for the description of thin films relies on the effective parameters being homogeneously distributed along the film thickness. Accounting for the magnetomechanical coupling makes the picture more complex, yet promises technological advantages: the cross-coupling between elastic, magnetic and electric subsystems opens additional degrees of freedom in the control of the respective order parameters[1–3]. The family of magnetomechanical effects includes piezo- and flexomagnetic responses, which determine the modification of the magnetic order parameter(s) due to

homogeneous or inhomogeneous strain, respectively. Piezomagnetic effects are well addressed in literature both for ferro- and antiferromagnetic materials including emergent anisotropy of material parameters when samples are exposed to uniaxial strain[4], change of material parameters in samples of different thickness[5], magnetoelasticity[6], transport effects in homogeneously strained films[7]. Despite the fact that flexomagnetism[8–13] is allowed for in the majority of bulk magnetic classes and enforced by boundaries[14], it is hardly addressed in the literature compared to the case of

[1]Helmholtz-Zentrum Dresden-Rossendorf e.V., Institute of Ion Beam Physics and Materials Research, 01328 Dresden, Germany. [2]Kyiv Academic University, 03142 Kyiv, Ukraine. [3]Department of Physics, University of Basel, 4056 Basel, Switzerland. [4]Institute for Solid State Research, Leibniz Institute for Solid State and Materials Research Dresden, 01069 Dresden, Germany. [5]Leibniz Institute for Solid State and Materials Research, IFW Dresden, Helmholtzstr. 20, 01069 Dresden, Germany. [6]Institute of Solid State and Materials Physics, TU Dresden, 01069 Dresden, Germany. [7]Helmholtz-Zentrum Dresden-Rossendorf e.V., Institute of Radiation Physics, 01328 Dresden, Germany. [8]These authors contributed equally: Pavlo Makushko, Tobias Kosub. ✉e-mail: o.pylypovskyi@hzdr.de; d.makarov@hzdr.de

piezomagnetism[15–19]. This situation is surprising considering that flexomagnetism is relevant for multiferroics and antiferromagnets, where magnetomechanical couplings lead to the control of the anisotropy and spontaneous magnetization[20–22]. Flexomagnetism can be found in magnets of 69 out of 90 bulk symmetry classes[14]. This includes the lowest symmetry groups 1 and $\bar{1}$ (54 nontrivial independent tensor components out of 81), which makes flexomagnetism to be readily expected even in Permalloy[10]. A higher symmetry of the lattice reduces number of independent components of the flexomagnetic tensor, e.g., leaving 10 independent components among 37 non-trivial ones for crystals of $\bar{3}'m'$ symmetry, or 5 independent components among non-trivial 18 ones for $\bar{6}'m2'$ crystals (e.g., $ThMn_2$). Lowering the symmetry at the surface allows the flexomagnetic effect in 19 surface magnetic classes, e.g., $(0\bar{1}10)$ surface cut of CrSb, which belongs to $6'/m'mm'$ bulk class[14].

The current understanding of the physics of flexomagnetism is limited to the prediction of the emergent ferromagnetic order parameter in the interior of an inhomogeneously strained antiferromagnetically ordered material. It is to be expected that the presence of the flexomagnetism-induced spontaneous ferromagnetic moment can provide novel but as-of-yet unexplored means for the manipulation of the antiferromagnetic order parameter, which could open new avenues for the design of non-volatile memory[23–25]. The fundamental understanding of the impact of flexomagnetism on thermodynamics of magnetic materials, as well as spin-reorientation and phase transitions, is missing at the moment. This lack in the fundamental knowledge of magnetomechanics in antiferromagnetic oxides hinders their application in prospective low-energy and high speed spintronic and spinorbitronic devices.

Here, we discover the presence of unconventional flexomagnetic effects in $Cr_2O_3$. Usually, flexomagnetism results in the magnetic moment induced by the strain gradient. For $Cr_2O_3$, we estimate this contribution to be 12 $\mu_B$ nm$^{-2}$ for the unit strain gradient. In contrast, we demonstrate another mechanism in the family of flexomagnetic effects, which originates in the impact of the strain gradient on thermodynamic properties, namely on the Néel temperature. The presence of the gradient of the Néel temperature across the thickness of $Cr_2O_3$ films results in an induced magnetic moment per unit strain gradient as high as 15 $\mu_B$ nm$^{-2}$. The distinct feature of this induced moment is that it follows the direction of the Néel vector. By combining magnetotransport and Nitrogen vacancy (NV) microscopy characterization data, we experimentally determine the presence of the gradient of the Néel temperature in 30-nm and 50-nm-thick $Cr_2O_3$ thin films. Accessing this magnetization is crucial for the proper description of the physics of antiferromagnetic solitons, including domain walls, skyrmions, and disclinations. In addition to these fundamental discoveries, we show that the inhomogeneously strained $Cr_2O_3$ thin films can have magnetic transition temperature of about 100 °C, which is about 60 °C higher than the transition temperature of the bulk $Cr_2O_3$. Due to good oxide-oxide heteroepitaxy, the Néel temperature in $Cr_2O_3$ thin films can be enhanced persistently and no sign of degradation of the Néel temperature is observed after 1.5 years. The physics discussed here is not specific to the $Cr_2O_3$. Based on the crystal symmetry and on the magnetic symmetry groups, this effect should also be observable in other relevant antiferromagnetic samples including CuMnAs and $Mn_2Au$. In this respect, the results of our work motivate further research on antiferromagnetic thin films, where magneto-mechanical coupling can play a key role in the design of material and device responses for ultrafast spintronics and magnonics[26–28] and applications in data storage[23,24,29].

## Results
### Inhomogeneously strained thin films
We prepared a series of $Cr_2O_3$ thin films of different thickness ranging from 30 to 250 nm on $(0\,0\,0\,1)$ oriented $Al_2O_3$ substrates using

magnetron sputtering (Supplementary Fig. 1). The samples are capped with a Pt layer to enable magnetotransport characterization (Fig. 1a). Regardless of the film thickness, $Cr_2O_3$ samples adopt the same orientation as the substrate and exhibit excellent crystallinity (Supplementary Figs. 2–8). Due to a wide deposition angle of the magnetron sputter source, the $Cr_2O_3$ films are dense and smooth (Supplementary Note 1, Supplementary Figs. 2–6). This is a major difference with respect to prior studies focused on thermally evaporated thin films[24,30], featuring columnar growth with clear grain boundaries going through the entire film thickness. The real space structure of the films is shown in the transmission electron microscopy (TEM) images in Fig. 1b,c for a 30-nm-thick film (Supplementary Figs. 4 and 5). A high-resolution TEM image (Fig. 1c) reveals that the misfit dislocations in such thin $Cr_2O_3$ films form several nanometers away from the interface with the $Al_2O_3$ substrate, leaving a significant amount of the film in a highly strained state that is pseudomorphic to $Al_2O_3$. The presence of the pseudomorphic $Cr_2O_3$ layer with a thickness of about 5 nm is also evidenced by the X-ray diffraction (XRD) characterization of 30- and 50-nm-thick samples (Supplementary Note 2, Supplementary Figs. 9–14). Thicker films do not reveal the presence of a pseudomorphic layer, but approach the bulk values of the lattice parameters (Supplementary Figs. 15–19).

Considering the absence of grain boundaries in the thin samples (Supplementary Note 1), we identify that the dominant strain relaxation mechanism in our samples is due to vacancy formation, their agglomeration and misfit dislocations. To access open volume defects, we perform the positron annihilation spectroscopy (PAS), revealing a change in the prevailing defect type from monovacancy-dislocation complexes to independent dislocations with the increase in the film thickness from 30 to 250 nm (Supplementary Note 3, Supplementary Figs. 20, 21). The size of the vacancy clusters at the top and bottom regions of the $Cr_2O_3$ film, which is linked to the measured positron annihilation time $\tau_2$ (Supplementary Fig. 22, Supplementary Table 1), decreases for thicker films suggesting decrease in strain (Fig. 1d). The size of agglomerated vacancies is found to be substantially smaller in the bottom part of the film compared to the top region. The difference between the vacancy cluster size at the top and bottom regions (nVac), normalized to the film thickness ($t$), can be used as a qualitative measure of the strain gradient in the film. These data are shown in Fig. 1e, suggesting a sizable strain gradient in 30-nm- and 50-nm-thick $Cr_2O_3$ films (Supplementary Fig. 23).

We utilize the $c/a$ ratio of the out-of-plane ($c$) and in-plane ($a$) lattice parameters as a measure of strain in $Cr_2O_3$ films. The lattice parameters averaged over the entire film thickness (Fig. 1f–h) are quantified relying on the analysis of the XRD data (Supplementary Note 2) and Raman spectroscopy characterization (Supplementary Note 4, Supplementary Figs. 24, 25). The thin film samples have developed compressive in-plane strain and show an extended c-axis. XRD data provide insights into the strain distribution along the thickness of the $Cr_2O_3$ film (Supplementary Note 2, Supplementary Figs. 10, 12–14, 16, 18). With increasing distance from the substrate, the strain gradually relaxes, resulting in a strain gradient along the film thickness, i.e. along the direction normal to the film interface (Fig. 1i, j, Supplementary Table 2 and Supplementary Fig. 26).

Strain modifies magnetic and thermodynamic properties of $Cr_2O_3$[16,19,31]. To evaluate the impact of the strain and strain gradient on the Néel temperature in thin films of sputter-deposited $Cr_2O_3$, we apply two complementary experimental methods. The Néel temperature at the top surface of the films, $T_N^{top}$, is assessed using magnetotransport measurements (Fig. 2). By monitoring the temperature evolution of the antiferromagnetic domain pattern using nitrogen vacancy (NV) scanning magnetometry, we evaluate the Néel temperature of the interior of the $Cr_2O_3$ films (Fig. 3).

## Magnetic state of the top surface

To assess the surface Néel temperature, we measure the transverse resistance of the Pt electrode ($R_{Tr}$) using the Zero-Offset Hall technique[24,32] (Supplementary Note 5). This property is modified by the magnetic state of the underlying $Cr_2O_3$ as a result of the transverse spin Hall magnetoresistance, which is a local probe of the magnetic ordering in the sample area adjacent to the Pt electrode[33–36]. Figure 2a, b shows the dependence of the transverse resistance on the out-of-plane applied magnetic field $H$, $R_{Tr}(H,T)$, for $Cr_2O_3$ thin film samples at selected temperatures (see also Supplementary Fig. 27): below (blue) and above (red) the transition temperature and in its vicinity (green). The $R_{Tr}(H,T)$ contains the

temperature dependent contribution from the antiferromagnetic $Cr_2O_3$ ($R_0(T)$, Supplementary Fig. 28), which we determine based on the $R_{Tr}(H,T)$ data (Fig. 2a). $R_0(T)$ is used for the evaluation of the Néel temperature at the top surface $T_N^{top}$ (Supplementary Note 5, Supplementary Figs. 29, 30), see Fig. 2c. For the 30-nm-thick film, $T_N^{top}$ is estimated to be higher than 75 °C (Fig. 2d). This is a major increase in the Néel temperature compared to the bulk value of $Cr_2O_3$, which is measured on a single crystal to be 35 °C (Fig. 2c, d). Thin films with a thickness of 250 nm reveal a Néel temperature similar to the single crystal value, confirming the relaxed state of the sample. The enhancement of the Néel temperature correlates with the average strain in the samples and is in line with earlier theoretical

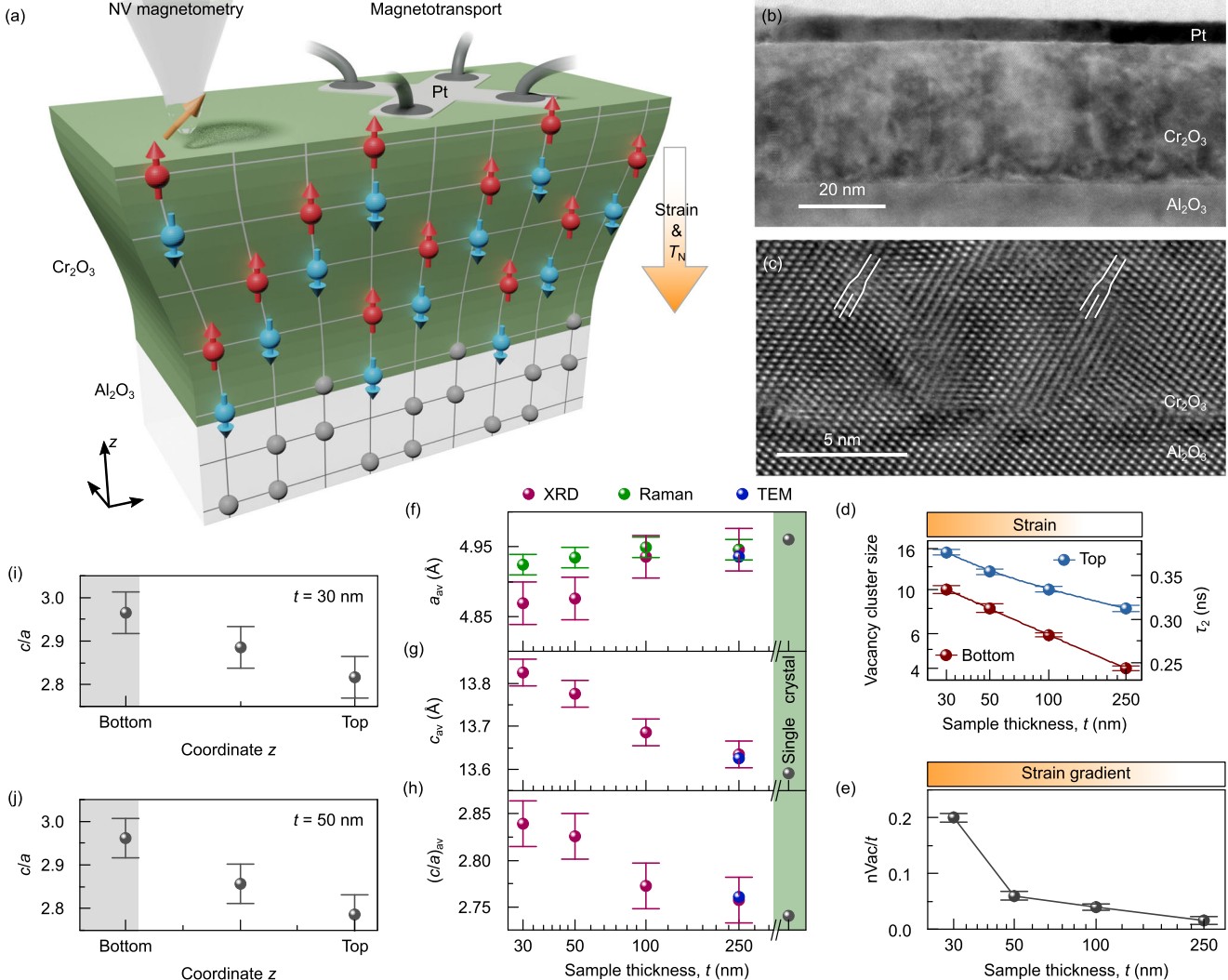

**Fig. 1 | Thin films of $Cr_2O_3$ with a strain gradient. a** Schematic of a $Cr_2O_3$ thin film prepared on a $Al_2O_3$ substrate. The film experiences a strain gradient along its thickness that leads to a gradient of the Néel temperature. The Néel temperature at the top surface is probed via magnetotransport relying on the transversal spin Hall magnetoresistance in a Pt electrode adjacent to the $Cr_2O_3$ film. The Néel temperature in the interior of the film is probed via NV magnetometry. **b** Cross-sectional bright-field TEM image showing the $Al_2O_3/Cr_2O_3/Pt$ stack for a 30-nm-thick sample. The sample does not reveal the presence of grain boundaries (see also Supplementary Note 1). **c** High-resolution TEM of the $Al_2O_3/Cr_2O_3$ interface. Two misfit dislocations are indicated in white. The dislocations are located about 5 nm away from the $Al_2O_3/Cr_2O_3$ interface. **d** Positron lifetime component $\tau_2$ and the corresponding calculated vacancy cluster size (assuming spherical geometry) in the top and bottom regions of the $Cr_2O_3$ films of different thickness $t$. The vacancy cluster size increases for thinner samples, indicating larger strain. Error bars

correspond to the standard deviations of the fit. **e** The gradient of the vacancy cluster size along the film thickness (nVac/$t$) for the samples of different thickness. This parameter is a qualitative measure of the strain gradient across the sample thickness. The error bars correspond to the standard deviations of the fit. Average values of the in-plane ($a$; (**f**)) and out-of-plane ($c$; (**g**)) lattice parameters and the respective $c/a$ ratio (**h**) estimated based on the XRD, Raman spectroscopy and TEM measurements of $Cr_2O_3$ samples of different thickness. Error bars of the lattice parameters obtained from XRD and Raman measurements are estimated from instrumental precision of the diffractometer and from the uncertainty of the Raman spectra peak fitting accordingly.(**I, j**) Change of the $c/a$ ratio along the film thickness estimated for the 30-nm-thick (**i**) and 50-nm-thick (**j**) samples. The $c/a$ ratio for the pseudomorpic layer is highlighted with the gray background. Error bars of the lattice parameters are estimated from instrumental precision of the diffractometer and from the uncertainty of the peak fitting.

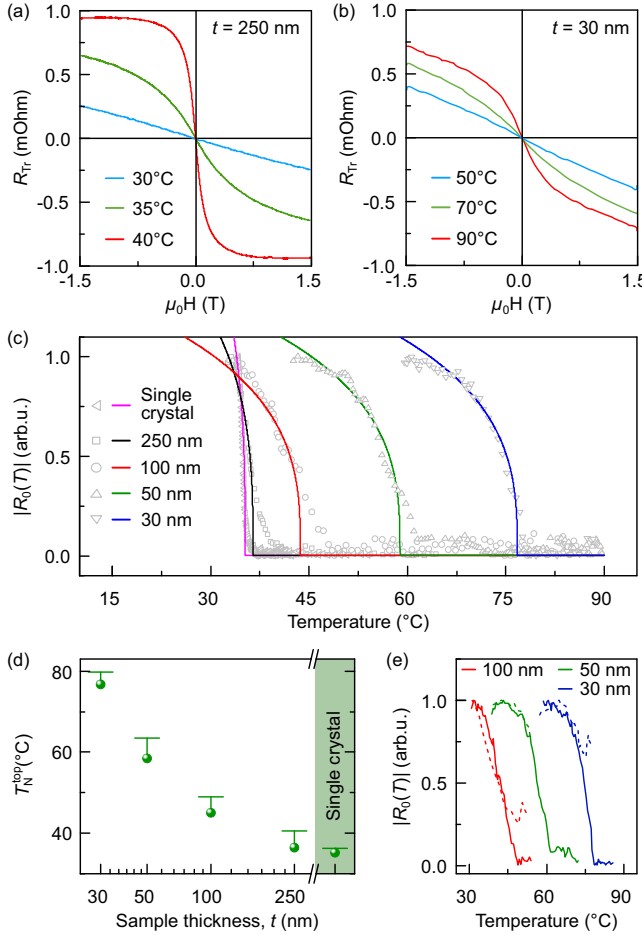

**Fig. 2 | Assessment of the Néel temperature $T_N^{top}$ at the top surface of Cr$_2$O$_3$ films. a, b** Transverse resistance $R_{Tr}$ of the Pt electrode vs out-of-plane magnetic field at temperatures below (blue), in the vicinity of (green) and above (red) the phase transition to the paramagnetic state for the samples of different thicknesses. **c** Temperature dependence of the antiferromagnetic contribution from the Cr$_2$O$_3$ (Supplementary Note 5) is shown with symbols. The fit used to determine the Néel temperature at the top surface of the sample is shown with solid lines. **d** Top surface Néel temperature $T_N^{top}$ vs sample thickness. The Néel temperature measured on a single crystal is shown as well. Error bars arise from the smooth decay of the $|R_0(T)|$ to the noise level in the vicinity of antiferromagnet-paramagnet transition. **e** Temperature evolution of the antiferromagnetic contribution from the Cr$_2$O$_3$ taken right after sample fabrication (dashed lines) and after 18 months (solid lines).

predictions[16,19,31]. Because the lateral strain relaxation is suppressed in sputter-deposited samples, they are persistently strained, assuring a long-term enhancement of the Néel temperature. This is evidenced by the close-to-identical magnetotransport responses taken directly following the sample fabrication and 18 months later (Fig. 2e, Supplementary Fig. 31).

## Magnetic state of the Cr$_2$O$_3$ films interior

We utilize NV magnetometry to assess the transition temperature within the interior of the films by following the temperature-induced decay of the magnetic contrast from the antiferromagnetic domain pattern, i.e., difference $\tilde{B}$ between the strength of stray fields measured within oppositely magnetized domains. This approach is similar to the detection of the Néel temperature using X-ray magnetic linear dichroism (XMLD) contrast in in-plane metallic antiferromagnets[37–40]. For the case of strain-free (250-nm-thick film, Supplementary Fig. 32) and strained (100-nm-thick film, Supplementary Fig. 33) samples, the

domain pattern in NV measurements disappears at temperatures similar to the Néel temperatures determined in magnetotransport studies, see Fig. 3a, b. We note that the surface and interior Néel temperatures assessed by magnetotransport and NV magnetometry do agree for the case of the strain-free Cr$_2$O$_3$ single crystal[29] and for thermally evaporated thin films[30]. In contrast, in thin-film samples with an inhomogeneous strain (50- and 30-nm-thick films, Supplementary Figs. 34, 35, respectively, see also Supplementary Figs. 5, 23 and Supplementary Note 1), the domain contrast remains clearly visible even at $T \gtrsim 85$ °C, see Fig. 3c, d. This is the highest temperature accessible in the setup. Thus, the Néel temperature measured with NV magnetometry is substantially larger than the respective value of $T_N^{top}$ (Supplementary Fig. 36). The observed difference in the transition temperature measured with the two methods is evidence for the gradient of the Néel temperature along the film thickness that arises from the inhomogeneous strain distribution (Fig. 1a, g, h, Supplementary Fig. 23).

## Flexomagnetism of Cr$_2$O$_3$ thin films

The discussion above suggests that gradient of the strain tensor $u_{ij}(z)$, where $z$ is the coordinate along the sample thickness, can impact antiferromagnetic materials not only via the emergence of ferromagnetism[14], but also by influencing the distribution of the transition temperature $T_N[u_{ij}(z)]$ along the thin film's thickness. This effect was not addressed before. For the phenomenological analysis, we assume that the strain varies linearly along the film thickness, $u_{ij} = u_{ij}^0 + w_{ij}z$ with $u_{ij}^0$ being the base level of strain and $w_{ij}$ being the strain gradient. In line with the study by Kota et al. [16] and multiscale modeling (Supplementary Note 6), this leads to a linear coordinate dependence of the Néel temperature $T_N(z) = T_N^{top} + \zeta \times (t - z)$, where $\zeta$ parameterizes the strain distribution in the film (Fig. 1i, j). In the vicinity of the antiferromagnet-paramagnet phase transition, the gradient of $T_N(z)$ enables a fundamentally new type of ferromagnetic contribution, which originates from the local variation of the length of the sublattice magnetization at $T =$ const, $\mathbf{M}_\zeta \propto \partial\mathbf{L}[T/T_N(z)]/\partial z$ (Supplementary Note 7, Supplementary Figs. 37, 38). This leads to the following form of the total flexomagnetic effect close to the phase transition $M = \mu \times \partial u/\partial z$, where $u = \langle u_{ij}\rangle$ is the average strain, $\mu = \mu^g + \mu^\zeta$ is the flexomagnetic coefficient. The first term, $\mu^g \sim 10$ $\mu_B$ nm$^{-2}$, originates from the gradient of $g$-tensor components due to the strain gradient (Supplementary Note 6, Supplementary Fig. 39). It leads to a constant magnetization along $z$ in 50- and 30-nm-thick Cr$_2$O$_3$ films, independent of the orientation of the Néel vector in antiferromagnetic domains. The second term, $\mu^\zeta$, is responsible for the appearance of $\mathbf{M}_\zeta$ with the spatial distribution following the order parameter $\mathbf{L}$ (Supplementary Note 7, Supplementary Fig. 38).

To perform quantitative estimations of the Néel temperature distribution and $\mu^\zeta$, we access the ferromagnetism of the sample (Supplementary Note 7) relying on the analysis of the strength of stray fields measured using NV magnetometry. The uncompensated surface spins (Supplementary Fig. 40) constitute the dominant contribution to the stray fields, and are linked to the direction of $\mathbf{L}$ in antiferromagnetic domains[29,30,41]. We model the strength of measured stray fields as a sum of contributions from the top and bottom film surfaces, complemented by $\mathbf{M}_\zeta$. The fit of the experimental data (Fig. 3b, d) then allows for an estimate of the transition temperature at the bottom interface, $T_N^{bot}$ (Supplementary Table 3). For the sample with a thickness of 100 nm, we find that $T_N^{bot} \approx T_N^{top}$. This is an expected behavior for thicker samples, as they do not reveal the presence of pseudomorphic layer or sizable strain gradient along the thickness. For the inhomogeneously strained film with the thickness of 50 nm, $T_N^{bot}$ is estimated to be about 100 °C, which is substantially higher than $T_N^{top}$ of 59 °C. This analysis leads to a value of $\mu^\zeta$ on the order of 15 $\mu_B$ nm$^{-2}$ at $T = 20$ °C.

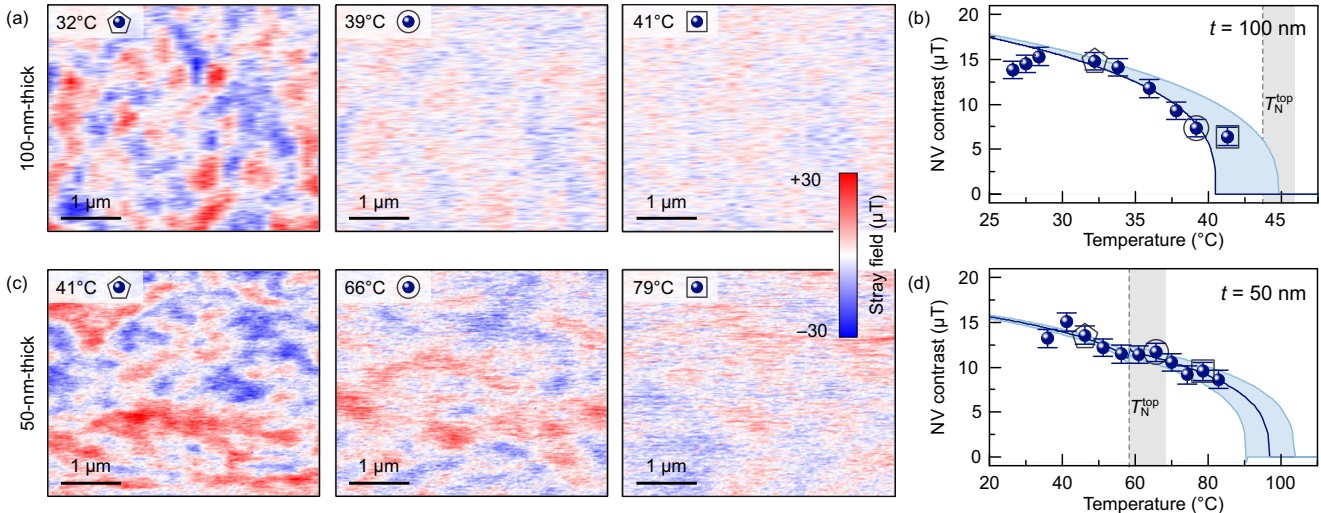

**Fig. 3 | Néel temperature of the interior of thin Cr₂O₃ films.** Temperature evolution of the antiferromagnetic domain pattern measured using NV magnetometry of (**a**) 100-nm-thick and (**c**) 50-nm-thick Cr₂O₃ samples. The strength of the measured stray fields decays upon approaching the transition temperature. Estimation of the phase transition temperature at the bottom interface, $T_N^{bot}$, of the (**b**) 100-nm-thick and (d) 50-nm-thick samples. The blue-shaded region in (**b** and **d**) corresponds to the uncertainty in the determination of the $T_N^{bot}$. The Néel temperature measured at the top surface via magnetotransport (Fig. 2) is shown for comparison in (**b** and **d**). A gray-shaded region indicates the uncertainty in the determination of the $T_N^{top}$. For the 50-nm-thick sample, a clear domain contrast is present even at temperature, which substantially higher than $T_N^{top}$. The strength of the stray field for scans shown in (**a** and **c**) are highlighted in (**b** and **d**) with corresponding symbols and error bars correspond to the half-width of the distribution.

## Vertically graded Néel temperature

The control of magnetic parameters of Cr₂O₃ via homogeneous stain (piezoeffect) is well established. In contrast, we demonstrate, that the flexomagnetic contribution is comparable in strength with the piezomagnetism and enhances even further the operation temperature of the devices based on technologically relevant Cr₂O₃ thin films. Due to the gradient of the Néel temperature along the sample thickness, different slices of the thin film experience the magnetic phase transition at different temperatures. Schematics of the phase transition for an inhomogeneously strained Cr₂O₃ film are shown in Fig. 4a–c. Far below $T_N^{top}$ (Fig. 4a), the length of the Néel vector $L$ is almost the same along the entire sample thickness (brown curve in Fig. 4d). Approaching $T_N^{top}$, the antiferromagnetic order parameter at the top surface becomes smaller and eventually vanishes at $T_N^{top}$ (Fig. 4b and red curve in Fig. 4d). At higher temperatures, the boundary between the antiferromagnetic and paramagnetic phases progresses towards the bottom surface of the Cr₂O₃ film (Fig. 4c and orange curve in Fig. 4d). The coexistence of para- and antiferromagnetic phases within the interior of the 50-nm-thick film is shown in Fig. 4e. We note that the continuously graded Néel temperature allows the magnetic nature of part of the film to be turned off and on.

Based on our results, the Néel temperature $T_N$ of inhomogeneously strained Cr₂O₃ thin films consists of the following contributions (Fig. 4f, Supplementary Note 7). The first term in $T_N$ is the bulk Néel temperature of Cr₂O₃ (green colored bar). The second contribution comes from the base level of strain $u_{ij}^0$ (orange colored bar). We note that to plot Fig. 4f, we set $u_{ij}^0$ to be equal to the smallest level of strain for each sample which is achieved at the top surface. This piezoeffect-driven increase in the Néel temperature determines $T_N^{top}$ and can be as high as 20 °C, in line with earlier theoretical predictions[16,19] (Supplementary Fig. 41). The third contribution is related to the increase of strain towards the bottom film surface (blue shaded bar refers to the maximal strain in the film). Due to the slow relaxation of strain along the film thickness, we see the pronounced and continuous increase in the Néel temperature towards the bottom interface. This results in the further increase in the transition temperature, reaching a remarkable 100 °C for 30- and 50-nm-thick Cr₂O₃ films. We note that despite

higher transition temperatures estimated for the 30-nm-thick film (Fig. 4f), the flexomagnetic effect in this sample can be reduced in comparison with the 50-nm-thick one because of lower gradient at smaller thickness.

## Discussion

In summary, we show that thin films of undoped Cr₂O₃ with inhomogeneously distributed strain possess a flexomagnetic response, which we probe by means of magnetotransport and NV scanning magnetometry measurements. The key enabler of the flexomagnetism is the proper design of the sample, revealing a persistent and pronounced strain gradient along the sample thickness, which is a result of (i) the chemical adhesion to the substrate due to the oxide-oxide heteroepitaxy and (ii) a grain boundary-free growth of thin films realized through a large magnetron sputter source. The latter prevents the formation of the interfacial misfit dislocations and an excessive strain relaxation into grain boundaries during the cool-down process. We note that the effect is not specific to Al₂O₃ substrates. Any substrate, which is isostructural to Cr₂O₃ (like V₂O₃) and can provide coherent epitaxy for Cr₂O₃ thin films should enable the discussed here flexomagnetic effect. In this respect, other substrates could offer further advantages. For instance, V₂O₃ is conducting and when growing Cr₂O₃ thin films can act as a gate enabling magnetoelectric characterization of the strained samples. Still, high miscibility of Cr₂O₃ and V₂O₃ should be be taken into account while preparing these samples.

From the fundamental point of view, our findings enable the experimental access to a subtle and still unexplored magnetomechanical coupling in thin films of antiferromagnetic insulator Cr₂O₃ but is also applicable for other antiferromagnetic materials including CuMnAs, Mn₂Au (magnetic symmetry group $mmm'$) and NiO (magnetic symmetry group $2/m'$)[14]. Although the $\bar{3}'m'$ bulk symmetry of the Cr₂O₃ and its boundary (0 0 0 1) allow for flexomagnetic effects[14], they have never been addressed for this material. Still, flexomagnetic effect is not generic. For instance, it is not expected in other well studied antiferromagnetic materials Mn₃Ir ($\bar{3}m'$ magnetic symmetry group) and MnF₂ ($4'/mmm'$ magnetic symmetry group)[14]. A spatial change of the material parameters along the thickness of thin films raises

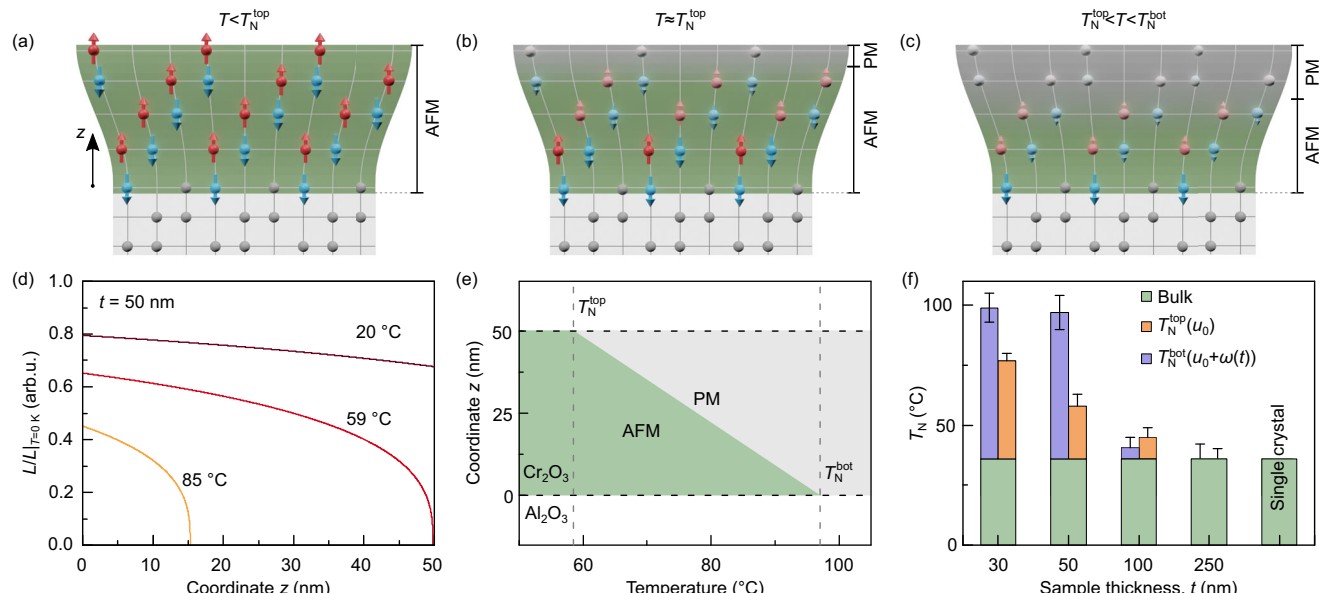

**Fig. 4 | Impact of the strain gradient on the antiferromagnet-paramagnet phase transition in Cr$_2$O$_3$ thin films. a–c** Schematics of the antiferromagnet-paramagnet phase transition. **a** At temperatures below $T_N^{top}$, the entire sample is anti-ferromagnetically ordered. **b** With the increase in temperature to about $T_N^{top}$, the less strained part of the sample (top region) undergoes a phase transition into the paramagnetic state. The bottom part of the sample remains antiferromagnetically ordered. **c** The portion of the sample in the ordered state is determined by the temperature. The sample turns into the paramagnetic state for $T > T_N^{bot}$. **d** The change of the length of the antiferromagnetic order parameter $L$ along the film thickness for a 50-nm-thick Cr$_2$O$_3$ thin film. The brown/red/orange curves correspond to the states shown in (**a–c**), respectively. **e** The portion of the 50-nm-thick film, which remains in the antiferromagnetic state, dependent on the measurement temperature. In the temperature range between $T_N^{top}$ and $T_N^{bot}$, only the part of the sample closer to the bottom interface remain in the ordered state. Green and gray shaded area represent regions in antiferromagnetic (AFM) and paramagnetic (PM) states, respectively. **f** Different contributions to the Néel temperature for Cr$_2$O$_3$ samples. The green shaded bar represents the bulk Néel temperature, which is measured for a single crystal and strain-free thin film (250-nm-thick in our study). The presence of a compressive strain in a Cr$_2$O$_3$ thin film leads to the increase in the Néel temperature. The value of $T_N$ at the top surface ($z = t$) which is determined by the smallest strain $u_0$ is indicated by the orange shaded bar. $T_N$ at the bottom surface (blue shaded bar) is larger according to the larger strain value $u_0 + wt$ at $z = 0$. Error bars on the $T_N^{top}$ and $T_N^{bot}$ correspond to uncertaininity of the fit of the magnetotransport and NV data, respectively.

questions about the applicability of the classical definition of the magnetic length as a universal characteristic of a magnetic thin film material. The community of high-resolution microscopies (X-ray- and electron-based techniques) can provide further unique insights into the spatially inhomogeneous antiferromagnet-paramagnet phase transition at the nanoscale, which can broaden our understanding of the physics of phase transitions in condensed matter. In addition to the spontaneous magnetization of flexomagnetic origin **M**, the symmetry of Cr$_2$O$_3$ allows a ferromagnetic moment to be produced by the gradient of the antiferromagnetic order parameter[14,42,43], for instance at the location of domain walls. This effect is not addressed in the present work, although we anticipate that NV magnetometry of thin films of Cr$_2$O$_3$ should be able to provide insight into this physics. Accessing this magnetization is crucial for the proper description of the physics of antiferromagnetic solitons, including domain walls, skyrmions, and disclinations[44,45]. Furthermore, our work poses questions on the magnetoelectric responses of these inhomogeneously strained thin films of Cr$_2$O$_3$. If confirmed experimentally, inhomogeneously strained magnetoelectric Cr$_2$O$_3$ thin films will become a game changer, enabling low-energy and high-speed spintronic and spin-orbitronic memory devices where magnetism is controlled by electric fields. The emergent flexomagnetism-driven ferromagnetic order parameter in antiferromagnetic thin films offers more flexibility in the design of spintronic and magnonic devices. In this respect, temperature becomes a useful tool for enabling the realization of reconfigurable racetracks and waveguides, given the unique possibility to deterministic turn off/on of the magnetism in less strained parts of the sample. For instance, a magnonic crystal can be designed as a stripe with modulated thickness. With the distribution of the Néel temperature along the stripe's normal, the relevant thickness of the magnetically

ordered interior of the sample can be controlled by temperature. The crystal's bandgap can be tuned, or its presence itself can be turned on or off by means of temperature. Furthermore, in contrast to statics, we anticipate that the magnetization of flexomagnetic origin should interact with spin waves in thin films due to the presence of stray fields[46–48]. Depending on the amplitude, it can be of relevance for spin Seebeck effect[49,50] and antiferromagnetic magnonics relying on antiferromagnetic insulators.

## Methods
### Sample preparation
Cr$_2$O$_3$ thin films were grown on *c*-cut single crystalline Al$_2$O$_3$(0001) substrates by RF magnetron sputtering (base pressure: below 10$^{-7}$ mbar; Ar sputter pressure: 10$^{-3}$ mbar; deposition rate: 1.72 nm min$^{-1}$; target size: 100 mm; sample to target separation: 10 cm) of a pure Cr$_2$O$_3$ target material (Robeko GmbH & Co. KG). Prior to the deposition of Cr$_2$O$_3$ the substrates were heated up to 750 °C for about 15 min to remove adsorbates from the surface and initiate the crystallization of Cr$_2$O$_3$ (Supplementary Fig. 1). Cr$_2$O$_3$ thin films of nominal thicknesses ranging from 30 to 250 nm were deposited at 700 °C. The high deposition temperature ensures a coherent epitaxial growth of Cr$_2$O$_3$ thin films. After the deposition, the substrate is cooled down by the heat dissipation through the sample mounting structure. The maximum cooling rate is 30 °C min$^{-1}$, resulting in a high level of thermally induced strain. To enable magnetotransport characterization, the samples were capped with a Pt layer in situ at room temperature (base pressure: 10$^{-7}$ mbar; Ar sputter pressure: 8 × 10$^{-4}$ mbar; deposition rate: 0.15 nm s$^{-1}$). A 1-mm-thick Cr$_2$O$_3$ single crystal (MaTecK, lattice parameters $a_0 = 4.96$ Å, $c_0 = 13.59$ Å) with (0 0 0 1) surface orientation was used as a reference sample. We note that the thickness of Pt capping

was different for $Cr_2O_3$ samples of different thickness: single crystal and 250-nm-thick $Cr_2O_3$ samples are capped with 3-nm-thick Pt; $Cr_2O_3$ films of other thickness are capped with 5-nm-thick Pt layer.

To address the absence of finite size effects in $Cr_2O_3$ samples with a thickness larger than 30 nm, we prepared an additional 15-nm-thick sample following the same growth protocol as discussed above. Due to the different criticality behavior (Supplementary Fig. 42), the 15-nm-thick $Cr_2O_3$ sample is not used for the analysis of the flexomagnetic effect and the flexomagnetism is discussed only for the samples with the thickness in the range between 30 and 250 nm.

The thickness of the $Cr_2O_3$ film affects the structural properties of the films due to a different strain state, which also affects the structure of the $Al_2O_3/Cr_2O_3$ interface. These two materials are isostructural with about 4% lattice mismatch within (0 0 0 1) plane. At the initial stages, $Cr_2O_3$ grows preudomorphically reproducing in-plane lattice parameters of the sapphire substrate. The pseudomorphic state is metastable. By forming this state, the chemical free energy competes with the strain free energy, which determines the critical thickness of the pseudomorphic layer[51]. During the growth, at some thickness of the film, it is energetically favorable to form dislocations, which are the limiting factor of the thickness of the pseudomorphic layer. Further, the film grows in non-pseudomorphic but highly strained state[52]. The location of the dislocation determines the thickness of the pseudomorphic layer (from the substrate to the plane with dislocations; Fig. 1 taken of a 30-nm-thick sample). With the increase in the total thickness of the $Cr_2O_3$ film, dislocations tend to diffuse towards the interface with the $Al_2O_3$ substrate. The pseudomorphic layer disappears when dislocations reach the interface with the substrate, while the coherent epitaxial relation of $Cr_2O_3$ and $Al_2O_3$ remains (Supplementary Fig. 2 taken of a 250-nm-thick sample).

The high-temperature high-field slope of $R_{Tr}$ in Supplementary Fig. 27 is strongly dependent on the thickness of the Pt electrode. The reduction of the thickness of the Pt layer resulted in a decrease of $\chi_{Pt}$ contribution to the measured transversal resistance. We note that the change in slope due to the normal Hall effect in Pt does not change the conclusion about the phase transition temperature. Indeed, for our analysis of the Néel temperature, we subtract this linear contribution as it is not dependent on the transition temperature of $Cr_2O_3$.

We checked that our thin films do not have ferromagnetic signal as was reported in[51,53–55] (Supplementary Fig. 43).

## Electron microscopy characterization

Cross-sectional bright-field and high-resolution TEM measurements were performed using an image-$C_s$-corrected Titan 80-300 microscope (FEI) operated at an accelerating voltage of 300 kV. High-angle annular dark-field scanning transmission electron microscopy (HAADF-STEM) imaging combined with spectrum imaging analysis based on energy-dispersive X-ray spectroscopy (EDX) were performed at 200 kV with a Talos F200X microscope equipped with an X-FEG electron source and a Super-X EDX detector system (FEI). Prior to (S)TEM analysis, the specimen was mounted in a double-tilt low-background holder and placed for 8 s into a Model 1020 Plasma Cleaner (Fischione) to remove possible contaminations. Classical TEM cross-sections, glued together in face-to-face geometry using G2 epoxy glue (Gatan), were prepared by sawing (Wire Saw WS 22, IBS GmbH), grinding (MetaServ 250, Bühler), polishing (Minimet 1000, Bühler), dimpling (Dimple Grinder 656, Gatan), and final Ar ion milling (Precision Ion Polishing System PIPS 691, Gatan). The TEM study was applied to understand the mechanism behind the persistent strain in magnetron-sputter-deposited $Cr_2O_3$ thin films and to confirm the high quality and coherent epitaxy of $Cr_2O_3$ thin films (Supplementary Note 1, Supplementary Fig. 2–6). TEM imaging revealed sharp and high-quality interfaces of $Cr_2O_3$ with the $Al_2O_3$ substrate and the Pt electrode. Fast Fourier transformation performed at the region of the $Cr_2O_3/Al_2O_3$ interface confirmed the

coherent epitaxial growth of the thin films on $Al_2O_3$ (Supplementary Figs. 3a, b and 5a, 6a, b for the 250-nm-thick and 30-nm-thick films, respectively). Furthermore, EDX analysis confirmed the phase purity of the thin films and the absence of intermixing at the $Cr_2O_3/Pt$ and $Cr_2O_3/Al_2O_3$ interfaces (Supplementary Figs. 3c, d and 6c, d for the 250-nm-thick and 30-nm-thick films, respectively).

High-resolution scanning TEM (HR-STEM) was performed on an FEI Titan[3] 80–300 probe-corrected instrument (ThermoFisher Company, US) operated at 300 kV. To this end, scan images were recorded with a high-angle annular dark-field (HAADF) detector at a camera length of 115 mm (collection angle 40–200 mrad) and a probe semi-convergence angle of 21.5 mrad.

## X-ray diffraction analysis

XRD studies were carried out using a Rigaku SmartLab 3 kW with a parallel beam of Cu-$K_{\alpha1}$ radiation, $\lambda = 1.54056$ Å (using a Ge (2 2 0) two-bounce monochromator). The reflection positions obtained by the profile deconvolution were corrected using the $Al_2O_3$ substrate. Lattice parameter refinement was performed with WinCSD[56].

## Raman spectroscopy

The Raman measurements were performed using a JY Horiba LabRAM HR Evolution Raman Spectrometer in the backscattering geometry. The excitation wavelength was 532 nm. The power of the incident beam was <1 mW. The radiation was focused and collected by a 100 × long working distance objective and the laser spot size was around 2 $\mu$m. The Rayleigh length for the beam divergence in the microscope is about 5 $\mu$m. As this is about 20 times larger than the thickness of our thickest film, we do not expect inhomogeneities along the film thickness. Besides multiple sharp peaks from the $Al_2O_3$ substrate, the Raman spectra of the $Cr_2O_3$ films are dominated by three relatively broad peaks (Supplementary Figs. 24, 25). The $Cr_2O_3$ spectra were fitted using a set of Lorentzian functions to obtain the phonon frequency dependence on the sample thickness. The variation of the lattice parameter has been estimated by comparing the observed phonon frequencies with literature data on pressure-induced phonon shift and the lattice compression[57]. All three phonon peaks demonstrate the frequency shift corresponding to the lattice compression. This agrees with the fact that the eigenvectors of the observed Raman-active phonons mainly involve the in-plane motion of the ions[58]. Correspondingly, they are mostly affected by the *in-plane compressive strain* of the $Cr_2O_3$ films. Using the XRD data on the strain-induced deformation and the pressure-induced shift of Raman modes[57], we estimate the variation of the averaged in-plane unit cell parameter $a_{av}$ with film thickness (Fig. 1f). We note that if the growth morphology results in a non-uniform strain, that produces a shoulder in the Raman peak. This behavior is well established for semiconductor nanostructures, see e.g. Bernardi et al. [59] for the case of Ge nanodots.

## Positron annihilation spectroscopy

Being implanted at a well-defined depth in the sample, positrons annihilate with electrons after localization in vacancy-like defects and their agglomerations (the so-called vacancy clusters or complexes; here voids) allowing for depth and concentration-dependent defect microstructure analysis. The positron annihilation lifetime experiments were carried out at the mono-energetic positron spectroscopy (MePS) beamline, the end station of the radiation source ELBE (Electron Linac for beams with high Brilliance and low Emittance) at the HZDR[60]. Signals were processed by a digital data acquisition system using a $CrBr_3$ scintillator detector coupled to a Hamamatsu R13089-100 photomultiplier tube. The data were analyzed using a custom software rooted at a SPDevices ADQ14DC-2X digitizer with 14 bit vertical resolution and 2 GS $s^{-1}$ horizontal resolution[61]. The overall measurement time resolution was not worse than 0.21 ns. A typical lifetime spectrum $N(t)$ is described by $N(t) = \sum(1/\tau_i)I_i \exp(-t/\tau_i)$, where $\tau_i$ and

$I_i$ are the positron lifetime and relative intensity of the $i$-th component, respectively ($\sum I_i = 1$). All the spectra were deconvoluted using the non-linearly least-squared based package PALSfit fitting software[62] into three discrete lifetime components, which provides evidence for two different defect types and a surface ortho-Positronium annihilation state. The second lifetime component $\tau_2$, representing vacancy agglomeration, is shown in Fig. 1d. The resolution function utilized for the spectrum analysis required two Gaussian functions with distinct intensities and shifts depending on the positron implantation energy $E_p$. All spectra contained at least $10^7$ counts. Further discussion on the positron annihilation spectroscopy studies is in Supplementary Note 3, see also Supplementary Fig. 20,21 and Supplementary Table 1. Theoretical calculations of positron lifetimes for the delocalized states (bulk lifetime) and for positrons trapped at defects states were obtained using the atomic superposition (ATSUP) method within two-component density functional theory (DFT) ab initio calculations[63]. For the electron-positron correlation, the generalized gradient approximation (GGA) scheme was used[64]. The calculated positron lifetimes corresponding to specific spherical-shaped void sizes (vacancy cluster sizes) are plotted in Supplementary Fig. 22.

## Magnetotransport measurements

Magnetotransport is a well-established technique to determine the phase transition temperature[65–68]. According to Wang et al. [69], the spin decay length in typical antiferromagnetic insulators including $Cr_2O_3$ is less than 10 nm (for $Cr_2O_3$ the spin decay length might be even smaller in the range of several nm only). This provides the upper bound on the depth from which we collect the information in the magnetotransport studies. Due to this surface sensitivity of the method, we refer to the transition temperature measured via magnetotransport as to the one measured at the top surface of the $Cr_2O_3$ film, $T_N^{top}$. We measured the magnetic field dependence of the transverse resistance, $R_{Tr}(H, T)$, of a Pt thin film deposited on $Cr_2O_3$ samples. The magnetotransport measurements were carried out using the Zero-Offset Hall preset of a Tensormeter measurement device[70], which provides the absolute transverse resistance without any offset due to undetermined sample geometry. Typical current amplitudes were on the order of 500 $\mu A$. For every sample, the $R_{Tr}(H, T)$ dependencies were measured for up to 100 different sample temperatures in the range from 15 °C up to 90 °C (this is the highest temperature, which is accessible in the setup), as shown in Fig. 2a, b and Supplementary Fig. 27. At low temperatures well in the antiferromagnetic state of the sample, the magnetic field dependence of the transversal resistance is a straight line. This behavior is given by the normal Hall effect of the Pt thin film (Supplementary Fig. 45). In the vicinity of the antiferromagnet-paramagnet transition temperature, the antiferromagnetic ordering is weakened and the contribution of the antiferromagnetic $Cr_2O_3$ to the measured transport signal becomes discernible (the slope of the $R_{Tr}(H, T)$ dependence taken at zero field increases). With further increase in the measurement temperature, the contribution from the $Cr_2O_3$ transforms the measured $R_{Tr}(H, T)$ dependence from a line to a Langevin-like function. By subtracting the normal Hall effect in Pt and the paramagnetic contribution from the $Cr_2O_3$ (Supplementary Note 5 and Supplementary Fig. 29), we access the temperature dependence of the contribution to the $R_{Tr}(H, T)$ from the antiferromagnetic $Cr_2O_3$. This is the $R_0(T)$ data shown in Fig. 2c. In a small temperature range close to the transition temperature, $R_0$ can be assumed to be proportional to the primary order parameter. Therefore, the dependencies $R_0(T)$ are used to determine the Néel temperature (Fig. 2c and Supplementary Fig. 29, 30, 42). Using this analysis method, we correctly capture the transition temperature of the single crystal to be 35 °C[71], see Supplementary Fig. 29a.

The $R_0(T)$ dependence for $Cr_2O_3$ thin films with the thickness of 30–250 nm can be described with the critical exponent $\beta \approx 0.3$, which is the same as for the reference $Cr_2O_3$ single crystal. This indicates the absence of finite size effects in these samples. On the contrary, a 15-nm-thick sample reveals the critical exponent $\beta$ of 0.67 (Supplementary Fig. 42), indicating the onset of finite size effects in this sample.

The magnetic polarization of Pt can be safely excluded from the consideration. In this respect, it is already established that Pt in proximity to antiferromagnetic insulators is not polarized[72–76]. This statement is also valid for $Cr_2O_3$/Pt interfaces, where it was experimentally confirmed that the measured transversal resistance is not an anomalous Hall effect but a spin-Hall magnetoresistance effect[35,77]. Here, we performed magnetotransport measurements of a 5-nm-thick Pt electrode deposited directly on a $Al_2O_3$ substrate (Supplementary Fig. 44). Besides a normal Hall effect due to paramagnetism of Pt, we do not observe any specific feature on the temperature and field dependence of the transversal resistance in the studied temperature range.

Alternatively, the change of the magnetic state at the $Cr_2O_3$/Pt interface can be detected by evaluating the change of the slope of the dependence $R_{Tr}(H=0, T)$. The transition of the $R_{Tr}(H=0, T)$ function from the linear to the Langevin-like shape upon the transition of the $Cr_2O_3$ to the paramagnetic state is accompanied by a pronounced change of the slope $\partial R_{Tr}/\partial H|_{H=0}$ at low (<100 mT) magnetic fields (see lines in Supplementary Fig. 45a). We analyze the variation of this parameter as a function of temperature according to $f_s(T) = \tanh T$ (Supplementary Fig. 45b). We correlate the transition temperature to the inflection point of the hyperbolic tangent function. The obtained values of the Néel temperature are summarized in Supplementary Fig. 45c. We note that the Néel temperatures estimated accordingly to these two methods of the analysis of the magnetotransport data are similar within the measurement error.

Growth of 30- and 250-nm-thick $Cr_2O_3$ films on a Pt seed layer leads to the relaxation of strain (Supplementary Figs. 46, 47, respectively) and low Néel temperature (Supplementary Fig. 30b).

## Nitrogen-vacancy magnetic microscopy

NV magnetometry makes use of the electronic spin triplet ground state of the atomic NV defect in diamond to measure stray fields at the nanoscale[78]. Scanning NV magnetometry was performed with a tip fabricated from single-crystal, ⟨100⟩-oriented diamond that was implanted with $^{14}N$ ions at 12 keV and annealed to form NV centers. A small bias field (<60 Oe) is applied along the NV axis (54° from the sample normal) to split the $m_s = |\pm 1\rangle$ states of the NV. The magnetic stray field along the NV axis is then proportional to the Zeeman splitting of the $m_s = |\pm 1\rangle$ states, allowing us to achieve quantitative nanoscale imaging. For imaging, a single NV center is located at the tip of an all-diamond scanning probe[79] to provide scanning magnetometry capabilities with ~50 nm spatial resolution. The imaging is performed using feedback to lock to one of the NV resonances[80], thereby extracting the local stray field. We note that NV magnetometry is sensitive to non-collinear magnetic textures only. The texture should provide magnetic stray fields, which are detected by the NV magnetometer. Thus, any homogeneous magnetization also the one coming from the saturated surface with uncompensated moments will not influence the measured signal. Furthermore, for the analysis of the $T_N^{bot}$ we follow the decay of the domain contrast

$$\widetilde{B} = B_{NV}^{red} - B_{NV}^{blue} \tag{1}$$

with $B_{NV}^{red}$ and $B_{NV}^{blue}$ being signed values of the average stray field strength measured on the oppositely ordered antiferromagnetic domains (red and blue areas in Fig. 3a, c). When the contrast (1) vanishes, we interpret this as the transition to the paramagnetic regime. No further assumptions are made for the determination of the Néel temperature in these measurements. This is determined by the detection limit of the setup, which provides $\widetilde{B} \sim 5 \mu T$ in the vicinity and above the phase transition. In this respect, the reported Néel

temperature is a lower bound and can be slightly higher. This means that the difference between the transport and NV estimates for the Néel temperature might be even higher, and the flexomagnetic contribution can be larger.

To build Fig. 3b, d and Supplementary Fig. 36, we average the measured stray field values from oppositely oriented domains separately. These average values were used to calculate $B_{NV}$. The measurements were carried out between room temperature and 85 °C, which is a hardware limitation for the setup. The temperature evolution of the antiferromagnetic domain pattern taken of the $Cr_2O_3$ samples of different thicknesses are shown in Supplementary Figs. 32–35. The NV contrast error bars of $2\,\mu T$ correspond to the halfwidth of the distribution.

Furthermore, we note that the presence of the Pt(3 nm) electrode does not affect the NV measurement of antiferromagnetic contrast $\widetilde{B}$ (Supplementary Fig. 48).

## Multiscale modeling

Molecular dynamics modeling of a $Cr_2O_3$ block containing 14,050 atoms ($60 \times 60 \times 100$ Å$^3$ simulation box) was done using the LAMMPS package[81] (Supplementary Note 6). The systems were parameterized with reactive force field Cr–Al–O[82]. We found that parameters for $Al_2O_3$ have a better agreement between theory and experiment. Therefore, instead of using the parameters of the $Cr_2O_3$ system and straining them down to the $Al_2O_3$ geometry at the bottom interface by 4%, we used parameters of $Al_2O_3$ and strained further by 4% to mimic the same relative geometric transformations. The structure optimization protocol includes quick NVT quenching down to $T = 10$ K with an overall time of 5000 fs following quick annealing at $T = 10$ K. An example of the LAMMPS protocol is provided in Supplementary Note 6. The first-principles CASSCF calculations were done at the DKH2/CAS(3,5)/RASSI-SO/ANO-RCC-MB level of theory using OpenMolcas[83]. In the complete active space (CAS) model, the total spin of the 3d-shell populated with three electrons is $S = 3/2$. Having solved the spin-free CAS problem, all 10 roots were used in further state interaction modeling using the spin–orbit coupling (SOC)-Hamiltonian. Finally, for all doubly degenerate SOC-states of the g-tensors were computed using first-order perturbation theory.

We used the D value fit in ruby as a standard for our ab initio anisotropy model in the distorted polyhedra of the strained $Cr_2O_3$. That is independent modeling for $CrO_6$-polyhedron as found in the $Al_2O_3$ structure. Then, we employed the ReaxFF model to predict a possible geometry of the asymmetrically loaded $Cr_2O_3$ fragment. This modeling produces a local polyhedra distortion along the c-axis and in the ab-pane. For each $CrO_6$-polyhedron in the predicted $Cr_2O_3$ structure, the D-value and $\mu_g$ was estimated. The limitations of the ReaxFF model are summarized in[82]. Also, for a given geometry, the local anisotropy can be predicted rather well, according to CASSCR/RASSI/SO literature[83].

We stress that the zero-field splitting is a function of all three lattice parameters for each Chromium ion and, in the general case, cannot be considered as a function of c value only. The parameter $c = c(z)$ is computed as the equilibrium value for the given strain and geometry. Thus, both distributions $D(z)$ and $c(z)$ reflect the actual state of the $Cr_2O_3$ epitaxially grown at $Al_2O_3$.

## Model of strained film

Typically, thin films are considered as homogeneous materials and variations of the micromagnetic parameters along the film thickness are neglected. In this approach, physical properties of epitaxial thin films (magnetic, electric, etc.) are linked to the average values of structural parameters like lattice constants or stress/strain state, which are often presented as functions of the overall film thickness. The primary focus of the manuscript is on the fact that thin film samples reveal variation of structural parameters along the normal direction of

the film, which results in peculiar magnetic effects. Only within this assumption that magnetic parameters change across the film thickness for the same sample, flexomagnetic effects can emerge in $Cr_2O_3$.

To describe inhomogeneously strained thin antiferromagnetic films (Supplementary Note 7), first we expand the Néel temperature into a power series as the function of the $c/a$ ratio. Having numerical evidence of a linear behavior of $T_N$ in a wide range of strain values[16] and considering thin films, we limit our description to the linear term only. In this way, we determine the expression (Supplementary Equation (S20)) for the position of the boundary between the paramagnetic and antiferromagnetically ordered regions of the thin film, which is used to calculate the curves in Fig. 4d,e. For quantitative analysis, we need to determine sources of the magnetic moment in the samples. We take into account the following contributions, which are written for the local density of magnetic moments per unit area of the film surface:

$$\sigma = \sigma_{surf} + \sigma_{dis} + \sigma_{flex} + \sigma_{d-l} + \sigma_{\zeta}. \tag{2}$$

(i) Uncompensated spins at the top and bottom surfaces[29,30,41] (Supplementary Fig. 40). This contribution gives the surface magnetization $\sigma_{surf}$ of about 2.2 $\mu_B$ nm$^{-2}$ for a single crystal[29] and a 200-nm-thick $Cr_2O_3$ film[30] at room temperature, which is several K below the transition temperature. We note that for a relaxed sample, the surface magnetization is the same at the top and bottom surfaces. For the following discussion, we define the direction of **L** to be associated with the direction of magnetic moments at the top surface.

(ii) Uncompensated spins at dislocations located close to the bottom film surface[24] (Supplementary Fig. 49). This contribution gives the surface magnetization of about $\sigma_{dis} = 0.45$ $\mu_B$ nm$^{-2}$ for a 200-nm-thick $Cr_2O_3$ film[24] close to the bottom interface with $Al_2O_3$. Recent studies carried out for $Cr_2O_3$ films of 250 nm thickness prepared under different protocols to reveal different defect nanostructure exhibit the Néel temperature close to single crystals[84]. Furthermore, even SPS-sintered polycrystalline $Cr_2O_3$ samples with random orientation of the easy axis across the volume of the sample reveal the bulk value of the Néel temperature of 307.5 ± 2 K[85]. This allows us to conclude that the local magnetic ordering and direction of the easy axis are not of importance for the determination of the Néel temperature.

(iii) Ferromagnetism induced by the strain gradient (the inverse flexomagnetic effect)[14] (Supplementary Fig. 50). The estimations of the strain gradient in our samples and flexomagnetic coefficient via multiscale modeling (Supplementary Note 6, Supplementary Fig. 39) gives the respective magnetic moment per unit area of the film of order of 0.7 $\mu_B$ nm$^{-2}$ (Supplementary Table 4).

(iv) Dzyaloshinskii-like coupling between the ferro- and antiferromagnetic order parameters at non-collinear textures results in $\sigma_{d-l} \neq 0$ (Supplementary Fig. 51). There are two mechanisms which lead to the appearance of magnetization with the effective surface density $\sigma_{d-l}$ induced by the gradients of the Néel vector in $Cr_2O_3$[42,43]. In the approach by Andreev [42], the spatial orientation of magnetization is determined by the spatial derivatives of **L** yet it is not necessarily aligned with **L**. The magnetization due to the flexoantiferromagnetic term introduced by Kabychenkov and Lisovskii [43] is mainly determined by the spatial derivatives of the in-plane components of **L**. The volume magnetization caused by this term is about 0.01 $\mu_B$ nm$^{-3}$ (see Supplementary Note 7).

In case of $Cr_2O_3$, the DMI links the antiferromagnetic order parameter $\mathbf{n} = \mathbf{m}_1 - \mathbf{m}_2 + \mathbf{m}_3 - \mathbf{m}_4$ with the secondary antiferromagnetic order parameter $\mathbf{n}_1 = \mathbf{m}_1 + \mathbf{m}_2 - \mathbf{m}_3 - \mathbf{m}_4$, where $\mathbf{m}_i$ denote magnetic moments of Cr ions within the unit cell[19,86]. This leads to the spin canted state with $\mathbf{m}_1 = -\mathbf{m}_4$ and $\mathbf{m}_2 = -\mathbf{m}_3$ resulting in a zero total magnetic moment. This situation is different for $\alpha$-$Fe_2O_3$, where $\mathbf{m}_1 = \mathbf{m}_4$ and $\mathbf{m}_2 = \mathbf{m}_3$ leads to the phenomenon of weak ferromagnetism[86]. Thus, even if spin canting is present in our samples, it cannot be detected via the NV magnetometry measurements.

(v) The spatially inhomogeneous reduction of the length of sub-lattice magnetizations (Supplementary Fig. 37) along the film thickness (and reduction of **L** as well) due to the inhomogeneous profile of the Néel temperature leads to the uncompensated magnetization

$$\mathbf{M}_\zeta \propto \frac{\mathbf{L}}{|\mathbf{L}|} \frac{\partial f(T/T_N)}{\partial z} \propto \frac{\partial u_{ij}}{\partial z}, \qquad (3)$$

where the function $f$ determines the temperature dependence of the sublattice magnetization and the Néel temperature is a function of strain tensor components $u_{ij}$. We estimate its strength using the temperature dependence of the sublattice magnetization. At $T = T_N^{top}$, the respective surface density $\sigma_\zeta = \int_0^t (\mathbf{M}_\zeta)_z \, dz \sim 2 \, \mu_B/nm^2$.

The strength of the aforementioned contributions is summarized in Supplementary Table 5. Piezomagnetism of $Cr_2O_3$ is not taken into account in our discussion, as it is forbidden in center-antisymmetric magnetic materials[87].

The inhomogeneous magnetic texture in the film plane leads to the appearance of stray fields, which can be detected using NV magnetometry. In our analysis, we rely on the numerical value of the NV contrast (1), which can be compared with the theoretical model we discuss in the following. The average stray field above the top surface reads

$$B_{NV} = \langle B_{top} \rangle + \langle B_{bot} \rangle + \langle B_\zeta \rangle. \qquad (4)$$

Here, $\langle \cdot \rangle$ means average over the NV scan area, $\langle B_{top} \rangle$ arises from the domains pattern at the top surface (contribution (i) discussed above), $\langle B_{bot} \rangle$ contains the signal from antiferromagnetic domains at the bottom film surface (contribution (i)) and contribution from dislocations (contribution (ii)), and $\langle B_\zeta \rangle$ contains the contribution from the magnetization (v)). It is reasonable to assume that defects-induced moments are randomly distributed in the film. Therefore, their explicit net contribution is averaged to zero. Hence, we cannot detect this contribution relying on the analysis of the NV contrast (1). In this way, we model the NV signal as a function of temperature as

$$B_{NV}(T) = \underbrace{b_1 \left[ 1 - \left( \frac{T}{T_N^{bot}} \right)^3 \right]^{1/3}}_{\text{bottom surface}} + \underbrace{b_2 \left[ 1 - \left( \frac{T}{T_N^{bot}} \right)^3 \right]^{1/3}}_{\text{top surface}}$$
$$+ \underbrace{b_3 \int_0^{t_{AFM}} |M_\zeta(T,z)| \, dz}_{\text{via reduction of L}}, \qquad (5)$$

where the parameters $b_{1,2,3}$ incorporate characteristics of the setup, $t_{AFM}$ is the thickness of the antiferromagnetic layer and $T_N^{bot} = T_N(0)$. The first two parameters $b_{1,2}$ contain information on the uncompensated surface magnetization; $b_3$ determines the average scale of the contribution from the interior of the film (Supplementary Fig. 52). There is a constraint on $b_1$ and $b_2$ that they should have opposite signs because they come from the oppositely oriented spins at the top and bottom surfaces[88]. The signal amplitude $b_3$ by definition consists of uncompensated sublattice magnetizations along the film thickness, which are scaling from the saturation magnetization of one sublattice to zero. Therefore, by magnitude, it should not exceed the signal from the bottom film surface where the magnetization length is maximal. We note that upon approaching the transition temperature, the magnetic length, which determines the domain wall width, diverges rendering the antiferromagnetic domains as effectively small ones. This explains why the stray field $B$ is measured not only at the domain walls,

but at the domains as well. In Equation (4), stray fields arising from contributions (iii) and (iv) are absent. The contribution (iii) induces magnetization with out-of-plane and in-plane components[14]. The out-of-plane component is homogeneous along the film thickness and, hence, it does not produce stray fields. The in-plane components are averaged to zero over the film area. The strength of the magnetization due to the mechanism (iv) is small and can be hardly detected in our measurements (Supplementary Note 7). We note that uncompensated moments located at point defects of arbitrary nature do not contribute to the value of (5) being averaged to zero.

Our theory is not directly applicable the region in the vicinity of $T_N$, where the renormalization group theory should be used to describe thermal fluctuations of magnetization and smooth transition from finite sublattice magnetization to the paramagnetic state. Taking the latter into account, will make the dependency $B_{NV}(T)$ a smoother function of temperature as well. We note that within our approach, the kinks in the temperature dependence of the NV contrast (Supplementary Fig. 36) is of the order of $2 \, \mu T$. Although irregularities in the NV contrast could be found in the relevant temperature range, the observed change is close to the resolution of our NV setup and is not discussed further.

The strength and direction of the magnetic moment, which is induced by the flexomagnetic effect is determined by the direction of the strain gradient with respect to crystallographic axes. In the absence of shear, the strain gradient along the $c$-axis can result in the magnetic moment, which is collinear with this axis. The strain gradient along the selected axis within the $c$ plane can result in the magnetic moment, inclined with respect to this axis, see Eliseev [14] and Eqns. (26) and (27) in Supplementary Notes of the manuscript. However, the thermally induced magnetic moment, $\mathbf{M}_\zeta$, is generated along the gradient of **L** only. The uncompensated magnetic moment of thermal origin is aligned along the Néel vector (see schematics in Supplementary Fig. 37).

## Data availability
The data supporting the findings of this study are available within the main text of this article and its Supplementary Information. Further information on this study is available from the corresponding authors upon reasonable request. Source data are provided with this paper.

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

## Acknowledgements

The authors thank Rainer Kaltofen, Dr. Ingolf Mönch, and Conrad Schubert (all HZDR) for their support with the sputter deposition of $Cr_2O_3$ thin films, Eoghan Cullen (HZDR) for the support in magnetotransport measurements, Dr. Jörg Grenzer and Andrea Scholz (both HZDR) for the support at initial stages of the project in the acquisition of the XRD data, and Romy Aniol for TEM specimen preparation. Support by the Ion Beam Center and ELBE large-scale facilities at the HZDR is gratefully acknowledged. R.H. acknowledges the funding of TEM Talos by the German Federal Ministry of Education and Research (BMBF), Grant No. 03SF0451, in the framework of HEMCP. A.L. and D.W. acknowledge funding from the European Research Council (ERC) under the Horizon 2020 research and innovation program of the European Union (grant agreement number 715620) and from the Deutsche Forschungsgemeinschaft (DFG, German Research Foundation, Project ID 431448015). This work was partially supported by the Impulse- und Networking fund of the Helmholtz Association (FKZ VH-VI-442 Memriox), and the Helmholtz Energy Materials Characterization Platform (03ET7015). This work was financed in part via the German Research Foundation (DFG) Grants No. MA5144/22-1, MA5144/24-1, and Helmholtz Association of German Research Centres in the frame of the Helmholtz Innovation Lab "FlexiSens".

## Author contributions

T.K. and P.Mak. fabricated and characterized the samples with the support of I.V. R.H., A.L., D.W. performed TEM characterization and data analysis. M.O.L., A.W., P.Mak. and I.V. collected and analyzed the PAS data. M.B. conducted positron DFT-ATSUP simulations. F.G. carried out XRD characterization with the support of I.V. N.H., K.W., B.J.S., P.L., and P.Mal. carried out and analyzed NV microscopy measurements with the support of O.V.P. and P.Mak. J.L. and A.P. carried out and analyzed Raman measurements. O.V.P. developed the phomenological model of flexomagnetism and applied it for the analysis of the experimental data with the support of P.Mak. and D.M. S.A. performed multiscale modeling to describe the strain propagation along the thickness of $Cr_2O_3$ films. P.Mak., T.K., O.V.P. and D.M. prepared the manuscript and organized figures. O.V.P. and D.M. developed the concept. J.F., P.Mal. and D.M. conceived the project. All authors contributed to discussions of the data and writing the manuscript.

## Funding

## Competing interests

The authors declare no competing interests.
