## [Peer Review File · Nature Communications]

Reviewers' Comments:

Reviewer #1:

Remarks to the Author:

The author answered my previous comments and questions carefully and discussed all the issues in detail. They have further clarified that flexomagnetism is caused by inhomogeneous strain originated from pseudomorphism in samples of specific thickness range. The manuscript now could better elaborate the distinction between piezomagnetism and the proposed flexomagnetism. Since the concept of flexomagnetism has not yet been widely discussed, some further questions should be clarified before further consideration.

1. According to the theory of flexomagnetism [Physical Review B 84, 174112 (2011)], only materials of few special point groups could obtain this effect. But the authors have cited [physica status solidi (RRL) – Rapid Research Letters 14, 1900467 (2019)], in which permalloy nanocrystals could be viewed as isotropic but also shows flexomagnetism. Since the authors have prospected more materials as candidates, it would be beneficial to comment on the symmetry requirements of this effect.
2. The antiferromagnetic order of Cr₂O₃ should be more carefully examined. It is known that easy-axis anisotropy of Cr₂O₃ might be broken due to nonperfect crystalline quality, for example in [Physical Review B 103, L020401 (2021)] there exists inclination of Néel vectors. The magnetic structure would strongly affect the measurement of Néel temperature.
3. It is mentioned that the 100 nm thick sample does not reveal a sizable strain gradient along the thickness, but the inhomogeneously strained film with the thickness of 50 nm reveals a sizable strained gradient, which leads to a value of μ on the order of $15 \mu_B/\text{nm}^2$. Is the absence of strain due to the strain relaxation during the growth from 50 to 100 nm or is it some uncontrollable result of material growth?
4. The authors mentioned that the flexomagnetic response arises from the inhomogeneity of the film. Can the magnitude of the flexomagnetism response be controlled by the inhomogeneity?
5. To obtain flexomagnetism, the method of fabrication is hardly repeatable and could be barely popularized. Would post growth annealing can relax the tension and eliminate flexomagnetism simultaneously? The authors could discuss further about the dynamics of thin film growth.

Reviewer #2:

Remarks to the Author:

This is a nice fundamental work which introduces novel flexomagnetic effects in an antiferromagnetic system (Cr₂O₃) – including z-dependent distributions of the Néel transition temperature and the antiferromagnetic (AFM) order parameter, which are accompanied by the emergence of flexomagnetism-induced moment. This finding is quite interesting and has the potential to initiate new explorations in antiferromagnetism. Authors do answer many questions raised by previous reviewers. That said, *a number of pressing scientific concerns/gaps have still not been addressed satisfactorily*. Moreover, some of the clarifications/data appended by the authors during the review *raise new questions that also need to be addressed*. Hence, the manuscript does not yet meet the standards for Nature Communications and must undergo major revision before it can be considered for publication.

1. Magnetotransport

- a. The temperature dependent evolution of χ_{Pt} is different in SI Fig 19b (black line) and SI Fig 34 (red line). Doesn't this imply that the presence of Cr₂O₃ has itself affected the thermal evolution of Pt? Authors should explain the origin of this effect. Moreover, the high-field (>1T) slopes of the $R_T(H)$ at highest temperatures (i.e. $T > T_{N^p}$), in SI Fig 18, are different across samples. According to Eqn S2 this should emerge from the Pt response as the paramagnetic Cr₂O₃ only makes a constant contribution (p_2). Why is the behaviour of Pt is changing so much across samples? Is this due to varying interface properties or surface quality (as remarked by Reviewer 3) or other reasons entirely? Authors identify all the ambiguities this varying Pt behaviour adds to the extraction of $R_0(H,T)$, if any.
- b. In SI Fig 35b, why are the slopes of $3R/3H$ so different as a function of sample thickness? This would imply that the AFM-PM transition occurs gradually over an increasingly broad range of temperatures as the sample thickness reduces. For instance, in the case of the 30 nm sample, the T_{N^p} transition is not complete even at 90 °C, putting it in the proximity of the $T_{N^{tot}}$. This *dramatically alters the estimated flexomagnetic effect* on T_N in this important sample. This information is not at all evident in main Fig 4f and related discussion. Authors should discuss the origin of this effect and explain if or how this changes the conclusions drawn from the magnetotransport data in SI Fig 20.
- c. Authors should clarify if the χ parameters of Pt and Cr₂O₃ given in Eqn S2 correspond to the typical magnetic susceptibility or the normal Hall effect (SI Note 5). Dimensional analysis doesn't seem to be consistent with the former, whereas an insulating Cr₂O₃ layer would not be able to contribute the latter. Hence, the notation is confusing. Why is χ_{Pt} negative in SI Fig 34b although the slopes are positive in SI Fig 34a? Is the paramagnetic behaviour of Pt expected to contribute to χ_{Pt} ?

- d. There are conflicting definitions of $R_{T_r,0}(H)$ in SI Note 5 and the caption of SI Fig 19d. The former eliminates, from $R_{T_r}(H)$, the contribution of Pt, whereas the latter subtracts the p_1 component of Cr_2O_3 . There also seems to be a mathematical error in the definition of $R_0(T)$ in the discussion after Eqn S4 (should the right side be $R_{T_r,0}$ or R_{T_r} ?). Authors need to correct such mathematical issues or else it can lead to misinterpretation of the data.
- e. Why is p_2 a constant for $T > T_{N_p}$ in Eqn S2? Authors should state if this an approximation, as this doesn't seem fully consistent with SI Fig 19c where p_2 increases systematically near and above T_{N_p} .

2. NV Imaging

- a. In response to Reviewer 1 authors state that “Upon fitting the NV data to Eq. (S17), we exclude the data points in the region where the contribution from the paramagnetic noise is dominant. Therefore, the paramagnetic noise does not influence the estimation of the T_N^{bot} .” This implies that authors would already have to ‘know’ where the paramagnetic region begins before performing the fitting which would then yield T_N , which seems like circular reasoning. This problem is exacerbated by the fact that the paramagnetic transition occurs at different temperatures across the sample, and no explicit term has been included in Eqn 4 to account for the NV signal contributed by paramagnetic regions in the upper parts of the sample for $T_{N_p} < T < T_N^{bot}$.
- b. 4 free parameters have been used to fit Eqn 4, of which 3 ($b_{1,2,3}$) depend on the setup (NV defect properties, defect-sample-distance etc.) and the magnetic structures in the sample (magnetization distribution in surface/bulk/interface regions, thickness of the respective regions etc.), which the authors know or control to a large degree. However, the authors have only stated rather simplistic fitting constraints in their rebuttal to Reviewer 1, which seems insufficient. In principle, the parameters $b_{1,2,3}$ could be determined by analysing the dipolar tensorial relationship between magnetization distribution and detected stray B-fields [see Nat Commun **6**, 7886 (2015), Nat Commun **9**, 2712 (2018), PRB **103**, 094426 (2021)]. Authors have performed such analysis in their previous work [Nat Phys **17**, 574 (2021)], and should include the relevant discussions in the SI/methods. As much as possible, they should *directly calculate or estimate* $b_{1,2,3}$, which can then be compared with the fitted values.

3. Crystal chemistry and structure

- a. There seems to be a gradual colour gradation of the pink contrast as a function of sample thickness in the EDX map of 250 nm sample, SI Fig 3d. The $Al_2O_3|Cr_2O_3$ interface seems to have a light contrast compared to $Cr_2O_3|Pt$ interface which is darker, with a clear dark blue hue (Cr-rich). This would imply that the top surface has more oxygen vacancies in comparison to the bottom one. Alternatively, the 30 nm sample has a homogenous pink contrast, SI Fig 5c, suggesting that the Cr/O ratio remains uniform across thickness. While this behaviour is not entirely surprising because the thicker samples stay at high temperatures for a prolonged period of time in an O-poor atmosphere during growth, it could *dramatically alter their magnetic properties* in comparison to thinner samples. It has been reported that O vacancies in Cr_2O_3 can result in ferromagnetism (FM) [see APL **105**, 132401 (2014); JAP **117**, 193907 (2015)], or modify AFM properties [see App Surf Sci **534**, 147638 (2020); APL **104**, 222402 (2014) and related references]. This aspect could *contribute to a completely different type of thickness-dependent AFM tuning in comparison to flexo-effects*, and even result in spurious FM signals. Hence, authors should carefully analyse the presence/evolution of oxygen vacancies across all samples. Moreover, they should explain the effect of this z-dependent chemical variation on the evolution of the Néel temperature and NV signals.
- b. Authors should perform systematic in-plane and out-of-plane SQUID magnetometry to rule out the presence of spurious FM contribution in all the samples.
- c. Addressing the question by Reviewers 1 and 2, authors cite the reference Acta Materialia **61**, 2703 (2013) and state that above a certain thickness of the layer, it is energetically favourable to induce dislocations at the top layer, which then diffuse into the interior of the film. While the first part of this sentence is agreeable, the second part is not that clear-cut. The cited reference mentions (on page 2709) that “Once the dislocations are nucleated at the free surface, their propagation to the film–substrate

interface is determined by the lattice frictional stress associated with dislocation glide in the film. In relatively soft materials, the propagation of dislocations is not the rate-limiting step for the relaxation process. However, in hard materials, such as III-nitrides and II-oxides, the propagation of dislocations plays a critical role by delaying the relaxation process.” This suggests that the dislocation propagation is a strongly material-dependent and temperature-dependent property. Moreover, the cited reference does not discuss any specific structural/strain evolution of Cr₂O₃ films on Al₂O₃ at all. Hence, the claim that bottommost layers in thicker Cr₂O₃ films (100 and 250 nm) are strain-free or pseudomorphic-layer-free still remains unsubstantiated.

- . Authors have added an HR-TEM image in SI Fig 2c, but its clarity is poorer in comparison to that of main Fig 1c or SI Fig 5a. Hence, it is difficult to judge the indicated positions of the misfit dislocations and absence/presence of a pseudomorphic layer.
- a. Authors assert in response to Reviewer 1 that RSM characterization is the most appropriate method to unambiguously determine thickness-dependent lattice parameters. However, the references provided [JAP **125**, 082508 (2019); APL **95**, 242902 (2009); AEM **4**, 1800106 (2018); Acta Materialia **58**, 6665 (2010)] primarily seem to use RSM analysis to determine either averaged lattice parameters of that sample or lattice parameters of multiple crystal fractions with clearly distinguishable and resolvable reciprocal space peaks. However, this is not at all the case with 1D multi-peak fitting deployed by the authors to resolve z-dependent sample fractions in the non-pseudomorphic regions, in SI Figs 8,10,12. It is evident from the figures below that the same line cut can be fitted in numerous ways (other than what is shown in SI Fig 8h), resulting in very different sample fractions and lattice constants. Moreover, surprisingly, the background of the peaks is not zero, in SI Figs 7-12, which adds further flexibility in the fitting process. In total for a fit involving double/triple peaks there are 7/10 free parameters, respectively. Also, authors do not discuss why they use 1D instead of the usual 2D Gaussian fitting of an RSM peak. Overall, despite the fact that the structural trends in SI Table 2 make general *qualitative* sense, authors’ current arguments have *failed to convince me* that the extensive (*semi*)*quantitative* fitting deployed in SI Figs 8,10,12 is clear or unambiguous.

- b. Given the foundational nature of this work in the antiferromagnetic community, structural experiments/analyses play an important role in estimating strain evolution and piezo- vs flexo-effects. Hence, I would have expected the authors to perform careful z-dependent lattice parameter quantification across all samples by other experimental means which are *direct* and *unambiguous*. One example is aberration-corrected scanning transmission electron microscopy in the high-angle annular dark field imaging (STEM-HAADF), which can provide atomic resolution and is much superior to HR-TEM. Such an approach has been nicely deployed in previous flexo-effect studies to characterize the z-dependent strain evolution [see Science **366**, 475 (2019)]. In principle, such an experiment would be straightforward to perform with the type of the samples in the present manuscript, and it would immediately put to rest all the concerns raised above in comments 3c-3e.

4. Modelling

- a. Authors have clarified in response to Reviewer 1 that the piezoeffect contribution is determined by the average strain. However, it is experimentally determined from the shift of the magnetotransport-derived $T_{N_{top}}$ relative to T_0 (see SI Note 7, SI Fig 32, SI Table 3). This does not seem correct as the strain experienced by the top surface is not the average strain, but the lowest strain (authors state this themselves in main Fig 4 caption). Based on the definition adopted by the authors, I would have expected the piezo-effect to be determined by $(T_{N_{top}} + T_{N_{bot}})/2$, for samples hosting linear decrease of T_N with thickness. Why is this not the case? Authors need to adopt a consistent framework tying theoretical and experimental analyses to avoid misinterpretation.
- b. SI Table 3 caption now states ΔT_{piezo} is estimated from Eqn S15 (DFT result), but the value given is from $T_{N_{top}} - T_0$ (experimental result). There is clearly some error here. Is there a mistake on the right most side of Eqn S14, as it seems to double count T_0 ?
- c. After Eqn S22, authors mention that $M_{fm,x}$ is averaged to zero across the film area. Does this mean that the sign and/or magnitude of this contribution would vary laterally across the sample? In principle, such variations could result in NV contrast if they are not too gradual. Hence, authors should discuss why these are not observed.

- d. Eqn S22 and S27 state that while $M_{fm,y}=0$, $M_{fm,x}\neq 0$ but weak. Is this due to crystallographic constraints in bulk Cr_2O_3 ? Are Cr_2O_3 thin films expected to have sizable in-plane anisotropy to result in such crystallographic constraints?

5. Other Comments:

- a. Authors use similar sounding terms across different contexts: e.g. NV contrast, stray fields, NV signal, $B_{NV} = B_{NV^{red}} - B_{NV^{blue}}$ and also $B_{NV} = \langle B_{top} \rangle + \langle B_{bot} \rangle + \langle B_{\zeta} \rangle$, in discussions/figures. Based on the discussion in the methods I surmise that they are all related, but they may imply different things. Other examples include the term μ_0 (permeability vs flexomagnetic coefficient), and the terms μ_z vs μ_{zz} . Authors should readjust the nomenclature, where possible, to avoid confusion.
- b. In the paragraph after Eqn S6, it is stated that μ_0 value has error of $\pm 4\%$, this is less than half the value given in Eqn S6. Definitions of the terms S , δ are not provided above Eqn S6.
- c. In SI Fig 8 the line cuts shown in panel c and d, do not seem to correspond clearly to the RSM map in panel b. For example, line cut F in panel c has a sharp peak near ~ 2.4 /nm, but this is not clearly seen along line cut F in panel b. Similar issues are also present in SI Fig S10. Is this an error in the captioning?
- d. Is there a typo in confidence intervals of the c and a lattice parameter values in SI Table 2? Also, c/a ratio error doesn't seem to follow standard error propagation. Do authors use a non-standard approach? Authors should include the data for 250 nm sample and bulk data in the table as well.
- e. The caption seems to be missing for SI Fig 40c?
- f. There are multiple occasions across the manuscript/SI/Figures, where the source of the error bars has not been explained.

We would like to thank the Reviewers for their remarks and suggestions, all of which are addressed in the revised manuscript. All changes in the manuscript are indicated in blue.

Our itemized responses to all the Reviewer's comments are below.

Reply to the Reviewer 1

Comment:

The author answered my previous comments and questions carefully and discussed all the issues in detail. They have further clarified that flexomagnetism is caused by inhomogeneous strain originated from pseudomorphism in samples of specific thickness range. The manuscript now could better elaborate the distinction between piezomagnetism and the proposed flexomagnetism. Since the concept of flexomagnetism has not yet been widely discussed, some further questions should be clarified before further consideration.

Answer:

We thank the Reviewer for his/her constructive remarks. We revised the manuscript accordingly and hope that the manuscript can be recommended for publication. The major changes to the manuscript are as follows:

1. We extended the discussion on flexomagnetic effects in samples of different symmetry groups.
2. We added the discussion on the impact of strain inhomogeneity on the flexomagnetism in Cr_2O_3 .
3. We commented on the impact of the crystal quality on the Neel temperature.

Comment 1:

According to the theory of flexomagnetism [Physical Review B 84, 174112 (2011)], only materials of few special point groups could obtain this effect. But the authors have cited [physica status solidi (RRL) – Rapid Research Letters 14, 1900467 (2019)], in which permalloy nanocrystals could be viewed as isotropic but also shows flexomagnetism. Since the authors have prospected more materials as candidates, it would be beneficial to comment on the symmetry requirements of this effect.

Answer:

We note that flexomagnetism can be found in magnets of 69 out of 90 bulk symmetry classes [1]. This includes the lowest symmetry groups 1 and $\bar{1}$ (54 nontrivial independent tensor components out of 81), which makes flexomagnetism to be readily expected in such alloys as $\text{Ni}_{71.5}\text{Fe}_{28.5}$ [2]. A higher symmetry of the lattice reduces the number of independent components of the flexomagnetic tensor, e.g., leaving 10 independent components among 37 non-trivial ones for crystals of $\bar{3}'m'$ symmetry, or 5 independent components among 18 ones for $\bar{6}'m2'$ crystals (e.g., ThMn_2). Lowering the symmetry at the surface allows the flexomagnetic effect in 19 surface magnetic classes, e.g., $(0\bar{1}10)$ surface cut of CrSb , which belongs to $6'/m'mm'$ bulk class [1].

We added this discussion in the introductory section of the manuscript (page 1).

Comment 2:

The antiferromagnetic order of Cr_2O_3 should be more carefully examined. It is known that easy axis anisotropy of Cr_2O_3 might be broken due to nonperfect crystalline quality, for example in [Physical Review B 103, L020401 (2021)] there exists inclination of Néel vectors. The magnetic structure would strongly affect the measurement of Néel temperature.

Answer:

We agree with the Reviewer that the crystal quality can affect the anisotropy of Cr_2O_3 . However, we do not expect that the randomization of the Neel vectors within the sample can influence its Neel temperature. For instance, since the discovery of magnetoelectricity in Cr_2O_3 , the Neel temperature of polycrystalline and single crystalline samples was found to be the same [3]. Recent studies carried out for Cr_2O_3 films of 250 nm thickness prepared under different

protocols to reveal different defect nanostructure exhibit the Néel temperature close to single crystals [4]. Furthermore, even SPS-sintered polycrystalline Cr_2O_3 samples with random orientation of the easy axis across the volume of the sample reveal the bulk value of the Néel temperature of 307.5 ± 2 K [3].

This remark is added to the revised manuscript (page 9).

Comment 3:

It is mentioned that the 100 nm thick sample does not reveal a sizable strain gradient along the thickness, but the inhomogeneously strained film with the thickness of 50 nm reveals a sizable strained gradient, which leads to a value of μ on the order of $15 \mu_{\text{B}}/\text{nm}^2$. Is the absence of strain due to the strain relaxation during the groweh from 50 to 100 nm or is it some uncontrollable result of material growth?

Answer:

The absence of the strain gradient in thicker samples is due to the strain relaxation on dislocations. They start to form in the films above the certain thickness and propagate to the bottom interface destroying the pseudomorphic layer (see Supplementary Note 1). PAS analysis suggests that the dislocation-vacancy complexes are dominant for films below 100 nm in thickness. For thicker films, free dislocations dominate (see Supplementary Note 3). As the samples are reproducible (repeated many times over last 4 years), we do not expect that the reported results are due to uncontrollable material growth.

Comment 4:

The authors mentioned that the flexomagnetic response arises from the inhomogeneity of the film. Can the magnitude of the flexomagnetism response be controlled by the inhomogeneity?

Answer:

Indeed, the flexomagnetic response can be controlled by inhomogeneities. Still, we would like to note that the inhomogeneity of the crystal lattice should be very specific because of the tensorial nature of the effect. For instance, the flexomagnetic tensor for $\bar{3}m'$ crystals has 10 non-zero independent components out of 81 ones (37 non-zero components) [1]. This allows to predict the total flexomagnetic response induced by strain along c axis (z axis in the laboratory reference frame) as

$$\begin{aligned} M_{\text{fm}x} &= -\mu_{xxxx} \left(\frac{\partial u_{xx}}{\partial z} - \frac{\partial u_{yy}}{\partial z} \right) - 2\mu_{xxzx} \frac{\partial u_{xz}}{\partial z}, \\ M_{\text{fm}y} &= 2\mu_{xxzx} \frac{\partial u_{xy}}{\partial z} - 2\mu_{xxzx} \frac{\partial u_{yz}}{\partial z}, \\ M_{\text{fm}z} &= -\mu_{xxzx} \left(\frac{\partial u_{xx}}{\partial z} + \frac{\partial u_{yy}}{\partial z} \right) - \mu_{zzzz} \frac{\partial u_{zz}}{\partial z}, \end{aligned} \quad (1)$$

In absence of shear ($u_{xz} \equiv u_{xy} \equiv u_{zx} \equiv 0$) and uniform in-plane strain, the magnetization is proportional to the coefficients μ_{xxzx} and μ_{zzzz} , and aligned with the c axis.

If the strain gradient exists along the x axis (e.g., for the Cr_2O_3 epitaxially grown in such a way to have c axis in the substrate plane and x axis is out-of-plane), the flexomagnetic response reads

$$\begin{aligned} M_{\text{fm}x} &= -\mu_{xxxx} \frac{\partial u_{xx}}{\partial x} - \mu_{xyxy} \frac{\partial u_{yy}}{\partial x} - \mu_{zzxx} \frac{\partial u_{zz}}{\partial x} - 2\mu_{xxzx} \frac{\partial u_{xz}}{\partial x}, \\ M_{\text{fm}y} &= (\mu_{xxzx} + \mu_{zzzx}) \frac{\partial u_{yz}}{\partial x} - 2\mu_{xyyx} \frac{\partial u_{xy}}{\partial x}, \\ M_{\text{fm}z} &= -2\mu_{xxzx} \frac{\partial u_{xz}}{\partial x} - \mu_{xxzx} \left(\frac{\partial u_{xx}}{\partial x} - \frac{\partial u_{yy}}{\partial x} \right). \end{aligned} \quad (2)$$

Unlike the case (1), even in absence of shear, there is the magnetization component aligned with the gradient ($M_{\text{fm}x}$) and along the c axis ($M_{\text{fm}z}$). They are determined by μ_{xxxx} , μ_{xyxy} , μ_{zzxx} , and by μ_{xxzx} , respectively. Thus, the magnitude and direction of the magnetization induced by flexomagnetic effect is strongly dependent on the way how the strain inhomogeneity is realized in the sample.

The respective discussion is added to the Supplementary Note 7D (page 15).

Comment 5:

To obtain flexomagnetism, the method of fabrication is hardly repeatable and could be barely popularized. Would post growth annealing can relax the tension and eliminate flexomagnetism simultaneously? The authors could discuss further about the dynamics of thin film growth.

Answer:

We kindly disagree with this comment of the Reviewer. The samples discussed in our work were grown using RF magnetron sputtering, which is a regular thin film fabrication tool readily available in many labs worldwide. We expect that when following our description on the sample preparation, the samples can be easily reproduced.

We anticipate that post growth annealing at high temperatures will relax strain present in Cr_2O_3 thin films. That would happen due to enhanced mobility of defects (vacancies, their clusters and dislocations), even despite that fact that the defects mobility is rather low in ceramic materials. We note that this study was not performed in the current work and requires further systematic investigations.

Reply to the Reviewer 2

Comment:

This is a nice fundamental work which introduces novel flexomagnetic effects in an antiferromagnetic system (Cr_2O_3) – including z-dependent distributions of the Néel transition temperature and the antiferromagnetic (AFM) order parameter, which are accompanied by the emergence of flexomagnetism-induced moment. This finding is quite interesting and has the potential to initiate new explorations in antiferromagnetism. Authors do answer many questions raised by previous reviewers. That said, *a number of pressing scientific concerns/gaps have still not been addressed satisfactorily*. Moreover, some of the clarifications/data appended by the authors during the review *raise new questions that also need to be addressed*. Hence, the manuscript does not yet meet the standards for Nature Communications and must undergo major revision before it can be considered for publication

Answer:

We thank the Reviewer for his/her insightful remarks, which stimulated us to carry out further experiments to support our claims. The new data is added to the manuscript (main text and supporting information). With these new discussions, the manuscript indeed became stronger and more complete.

The major changes to the manuscript are as follows:

1. We provide additional SQUID-VSM characterization indicating that our samples do not reveal magnetic moment.
2. We performed HR-STEM measurements on the sample with the thickness of 30 nm of Cr_2O_3 revealing the strain gradient across the film thickness.
3. We performed a new set of XRD measurements to support our claims on the strain state in the samples of different thickness.

Comment 1:

Magnetotransport

Comment 1a-1:

The temperature dependent evolution of χ_{Pt} is different in SI Fig 19b (black line) and SI Fig 34 (red line). Doesn't this imply that the presence of Cr_2O_3 has itself affected the thermal evolution of Pt? Authors should explain the origin of this effect.

Answer:

Indeed, the presence of Cr_2O_3 in the stack in addition to the Pt thin film necessarily affects the thermal evolution of the Hall signal measured using Pt top electrode [5–7]. In the stack $\text{Cr}_2\text{O}_3/\text{Pt}$, we measure the spin Hall effect as function of temperature using Pt electrodes. The Hall resistance measured on Pt is modified by the underlying Cr_2O_3 due to the finite spin decay length in Cr_2O_3 , which allows to alter Pt conductance by the magnetic ordering of Cr_2O_3 .

Furthermore, we note that the samples have different Pt layer thickness. The samples with the Cr_2O_3 layer with the thickness of 15, 30, 50, 100 nm are capped with a 5-nm-thick Pt layer. The 250-nm-thick Cr_2O_3 film has Pt(3 nm). In the two figures, mentioned by the Reviewer, the thickness of Pt is different (3-nm-thick in SI Fig. 19b (SI Fig. 28 after revision); 5-nm-thick in SI Fig. 34 (SI Fig. 44 after revision)). This explains the difference in the slope for the normal Hall effect.

The different thickness of Pt layer does not affect the results of our analysis on the Neel temperature. In the $\text{Cr}_2\text{O}_3/\text{Pt}$ system, the measured transversal signal consists of the normal Hall effect of platinum and spin Hall effect of the stack $\text{Cr}_2\text{O}_3/\text{Pt}$. The contribution from the normal Hall effect in Pt is subtracted from the data measured for each sample individually. The subtracted dependencies of the normal Hall were not used for further analysis.

This discussion is added to the revised manuscript (Supplementary Note 5, page 8-9).

Comment 1a-2:

Moreover, the high-field ($>1\text{T}$) slopes of the $R_{\text{Tr}}(\text{H})$ at highest temperatures (i.e. $T > T_{\text{N}}^{\text{top}}$), in SI Fig 18, are different across samples. According to Eqn S2 this should emerge from the Pt response as the paramagnetic Cr_2O_3 only makes a constant contribution (p_2). Why is the behaviour of Pt is changing

so much across samples? Is this due to varying interface properties or surface quality (as remarked by Reviewer 3) or other reasons entirely? Authors identify all the ambiguities this varying Pt behaviour adds to the extraction of $R_0(H,T)$, if any.

Answer:

The high-temperature high-field slope of R_{Tr} in SI Fig. 18 (SI Fig. 27 after revision) is strongly dependent on the thickness of the Pt electrode (single crystal and 250-nm-thick Cr_2O_3 samples are capped with 3-nm-thick Pt; Cr_2O_3 films of other thickness are capped with 5-nm-thick Pt layer). The reduction of the thickness of the Pt layer resulted in a decrease of χ_{Pt} contribution to the measured transversal resistance. We note that the slope due to normal Hall effect in Pt does not change the conclusion about the phase transition temperature. Indeed, for our analysis of the Neel temperature, we subtract this linear contribution as it is not dependent on the transition temperature of Cr_2O_3 .

The remark about Pt thickness is added to the Supplementary Note 5 (page 9) and to the section on the sample fabrication (page 5).

Comment 1b:

In SI Fig 35b, why are the slopes of $\partial R/\partial H$ so different as a function of sample thickness? This would imply that the AFM-PM transition occurs gradually over an increasingly broad range of temperatures as the sample thickness reduces. For instance, in the case of the 30 nm sample, the T_N^{top} transition is not complete even at 90 °C, putting it in the proximity of the T_N^{bot} . This *dramatically alters the estimated flexomagnetic effect* on T_N in this important sample. This information is not at all evident in main Fig 4f and related discussion. Authors should discuss the origin of this effect and explain if or how this changes the conclusions drawn from the magnetotransport data in SI Fig 20.

Answer:

The Reviewer is right, the transition temperature gradually changes with the sample's thickness. This behavior is well-known for thin films [5, 8]. The mechanism, which is discussed in literature is due to the distribution of temperatures for each grain in polycrystalline Cr_2O_3 films. Furthermore, with the decrease of the film thickness, the criticality changes and the order parameter decays with the temperature linearly.

The 30-nm-thick-sample would be very important for discussion. However, we do not have enough temperature range to assess its transition temperature. This information is already available in the manuscript. Therefore, our *quantitative* analysis is primarily based on the 50-nm-thick sample. With regard to the 30-nm-thick sample, we can only state that the transition temperature can be underestimated. This remark is now added in the paragraph just before the discussion section (page 4).

We note that the Supplementary Fig. 44c shows that there is no substantial difference in the value of the transition temperature T_N^{top} obtained by different approaches to the analysis of the Hall measurements.

Comment 1c:

Authors should clarify if the χ parameters of Pt and Cr_2O_3 given in Eqn S2 correspond to the typical magnetic susceptibility or the normal Hall effect (SI Note 5). Dimensional analysis doesn't seem to be consistent with the former, whereas an insulating Cr_2O_3 layer would not be able to contribute the latter. Hence, the notation is confusing. Why is χ_{Pt} negative in SI Fig 34b although the slopes are positive in SI Fig 34a? Is the paramagnetic behaviour of Pt expected to contribute to χ_{Pt} ?

Answer:

The Reviewer is right, χ_{Pt} is not the magnetic susceptibility. The respective sentence was misleadingly formulated. This is a coefficient, which has units of Ohm/T and characterizes the change of the transversal resistance of Pt with temperature and magnetic field. We revised the text to avoid misinterpretations.

χ_{Pt} contains paramagnetic signal from the Pt layer, which grows linearly with magnetic field within the given temperature range. This behavior is clearly seen in slopes of R_{Tr} in Supplementary Fig. 27c–e.

We corrected the notation for χ_{Pt} (page 8 of Supplementary Information).

We thank the Reviewer for noting this inconsistency in the presentation of the slope in SI Fig. 34b (SI Fig. 44 after revision). This is now corrected in the revised version.

Comment 1d:

There are conflicting definitions of $R_{Tr,0}(H)$ in SI Note 5 and the caption of SI Fig 19d. The former eliminates, from $R_{Tr}(H)$, the contribution of Pt, whereas the latter subtracts the p_1 component of Cr_2O_3 . There also seems to be a mathematical error in the definition of $R_0(T)$ in the discussion after Eqn S4 (should the right side be $R_{Tr,0}$ or R_{Tr} ?). Authors need to correct such mathematical issues or else it can lead to misinterpretation of the data.

Answer:

There was an inconsistency in the caption of SI Fig. 19d (SI Fig. 28 after revision). To improve clarity, we adjusted related explanations in the Supplementary Note 5 (pages 8–9). We described the step-by-step analysis of the magnetotransport data and replaced equations in the figure caption by references to the SI Note 5.

Comment 1e:

Why is p_2 a constant for $T > T_N^{top}$ in Eqn S2? Authors should state if this an approximation, as this doesn't seem fully consistent with SI Fig 19c where p_2 increases systematically near and above T_N^{top} .

Answer:

The Reviewer is right. Indeed, the parameter p_2 increases above the Neel temperature. This can be related to the grows of the magnetic susceptibility of Cr_2O_3 above T_N , e.g., see Fig. 8 in [9] or Fig. 3a in [3].

In this respect, the statement $p_2 = \text{const}$ is an approximation of our model. This parameter includes all contributions to the Hall resistance, which are non-linear with respect to the magnetic field H and are not substantial for the determination of the phase transition temperature of Cr_2O_3 . Furthermore, the model can be improved to take into account correlations at higher temperatures. However, building a full microscopic model for this process is a challenging task and should be done separately.

We added the respective remark to the Supplementary Note 5 (page 8).

Comment 2:**NV Imaging****Comment 2a-1:**

In response to Reviewer 1 authors state that “Upon fitting the NV data to Eq. (S17), we exclude the data points in the region where the contribution from the paramagnetic noise is dominant. Therefore, the paramagnetic noise does not influence the estimation of the T_N^{bot} .” This implies that authors would already have to ‘know’ where the paramagnetic region begins before performing the fitting which would then yield T_N , which seems like circular reasoning.

Answer:

In our analysis, we exclude the images, which do not contain a clear domain pattern. This can be the case if the sample is in the paramagnetic state or the signal is below the noise of the experimental setup. In this respect, NV data provides an estimation of the lower boundary for T_N^{bot} and, therefore, gives the lower boundary for the flexomagnetic effect. This remark was already provided in the manuscript (page 8).

Comment 2a-2:

This problem is exacerbated by the fact that the paramagnetic transition occurs at different temperatures across the sample, and no explicit term has been included in Eqn 4 to account for the NV signal contributed by paramagnetic regions in the upper parts of the sample for $T_N^{top} < T < T_N^{bot}$.

Answer:

The amplitude of the NV contrast \tilde{B} measured of the sample in the paramagnetic state is $2.5 \mu\text{T}$ providing the detection limit of about $5 \mu\text{T}$ (SI Figures 32i and 33k) with zero average of $B(\mathbf{r})$ across the film. For the 50-nm-thick film, the NV contrast is $13.5 \mu\text{T}$ (SI Figure 34l). Since we define the NV signal measured at domains of the given

color as the average over all respective pixels, the presence of the paramagnetic contribution does not influence this value.

Comment 2b:

4 free parameters have been used to fit Eqn 4, of which 3 ($b_{1,2,3}$) depend on the setup (NV defect properties, defect-sample-distance etc.) and the magnetic structures in the sample (magnetization distribution in surface/bulk/interface regions, thickness of the respective regions etc.), which the authors know or control to a large degree. However, the authors have only stated rather simplistic fitting constraints in their rebuttal to Reviewer 1, which seems insufficient. In principle, the parameters $b_{1,2,3}$ could be determined by analysing the dipolar tensorial relationship between magnetization distribution and detected stray B-fields [see Nat Commun 6, 7886 (2015), Nat Commun 9, 2712 (2018), PRB 103, 094426 (2021)]. Authors have performed such analysis in their previous work [Nat Phys 17, 574 (2021)], and should include the relevant discussions in the SI/methods. As much as possible, they should *directly calculate or estimate* $b_{1,2,3}$, which can then be compared with the fitted values.

Answer:

We note that using of four and even more fitting parameters for NV data is a typical approach [8, 10–12] because recovering of the magnetization distribution from the stray fields needs an initial guess about the stray field source [13] such as topography of the sample’s surface [8] or magnetic texture defined via set of free parameters as its source (e.g., domain wall width and angle of the order parameter within the domain wall) [8, 10, 11]. The statement of the Reviewer that the magnetic structures in the samples are known on the level to be directly utilized for stray field mapping is not justified because the NV imaging is an indirect method to measure magnetization. The latter is necessary for our approach to access the interior of the film to address thermally induced flexomagnetism. Namely, the Fourier map of stray fields integrated from the sample of thickness t reads

$$B(\mathbf{k}) = \int_0^t B_0(\mathbf{k}, z) k e^{-k(d+t-z)} dz, \quad (3)$$

$$B_0(\mathbf{k}, z) = \frac{\mu_0 M_s(z)}{2} \begin{pmatrix} -\frac{k_x^2}{k^2} & -\frac{1}{2} \sin\left(2 \arctan \frac{k_y}{k_x}\right) & -\frac{ik_x}{k} \\ -\frac{1}{2} \sin\left(2 \arctan \frac{k_y}{k_x}\right) & -\frac{k_y^2}{k^2} & -\frac{ik_y}{k} \\ -\frac{ik_x}{k} & -\frac{ik_y}{k} & 1 \end{pmatrix} \times \begin{pmatrix} m_x(\mathbf{k}, z) \\ m_y(\mathbf{k}, z) \\ m_z(\mathbf{k}, z) \end{pmatrix},$$

where $\mathbf{k} = \{k_x, k_y, 0\}$ with $k = |\mathbf{k}|$ is the wave vector of the Fourier transform, d is the distance from the top surface to the NV tip, $M_s(z)$ and $\mathbf{m}(\mathbf{k}, z) = \{m_x, m_y, m_z\}$ are the saturation magnetization and the unit vector characterizing the magnetic texture at the distance z from the bottom film interface, respectively. The exact shape of $\mathbf{m}(\mathbf{k}, z)$ is not known in contrast to the problems, where the stray fields of a single skyrmion or a domain wall are considered. That’s why we use an approximation stated as Eq. (5), which contains the key contributions to the measured value of stray fields.

To unambiguously solve the inverse problem of finding the magnetization from the measured stray fields distribution above the sample’s surface, one needs to access a 3D distribution of the Néel order parameter at different temperatures, which is not established for antiferromagnets to the best of our knowledge. Furthermore, being available, such a technique could be able to provide a direct measurement of both, T_N^{top} and $T_N^{\text{bot}} = T_N^{\text{top}} + \zeta t$ by a sequence of measurements of 3D domain structure. In this case, the gradient of the Néel temperature and thermal contribution to the flexomagnetic effect can be accessed directly without the need of combination of magnetotransport and NV magnetometry.

Due to the aforementioned challenges, a local spatial mapping of the stay fields distribution obtained from NV measurements to the magnetization distribution (as suggested by the Reviewer) requires many more hardly justifiable guesses and parameters than those 4 used in our manuscript. That was the reason to treat the coefficients $b_{1,2,3}$ as phenomenological. We are convinced that an independent estimation of the coefficients $b_{1,2,3}$ is hardly possible and have been done neither by us nor other groups.

This discussion is added to the revised manuscript (Supplementary Information, page 18).

Comment 3:
Crystal chemistry and structure

Comment 3a:

There seems to be a gradual colour gradation of the pink contrast as a function of sample thickness in the EDX map of 250 nm sample, SI Fig 3d. The $\text{Al}_2\text{O}_3|\text{Cr}_2\text{O}_3$ interface seems to have a light contrast compared to $\text{Cr}_2\text{O}_3|\text{Pt}$ interface which is darker, with a clear dark blue hue (Cr-rich). This would imply that the top surface has more oxygen vacancies in comparison to the bottom one. Alternatively, the 30 nm sample has a homogenous pink contrast, SI Fig 5c, suggesting that the Cr/O ratio remains uniform across thickness. While this behaviour is not entirely surprising because the thicker samples stay at high temperatures for a prolonged period of time in an O-poor atmosphere during growth, it could *dramatically alter their magnetic properties* in comparison to thinner samples. It has been reported that O vacancies in Cr_2O_3 can result in ferromagnetism (FM) [see APL 105, 132401 (2014); JAP 117, 193907 (2015)], or modify AFM properties [see App Surf Sci 534, 147638 (2020); APL 104, 222402 (2014) and related references]. This aspect could *contribute to a completely different type of thickness-dependent AFM tuning in comparison to flexo-effects*, and even result in spurious FM signals. Hence, authors should carefully analyse the presence/evolution of oxygen vacancies across all samples. Moreover, they should explain the effect of this z-dependent chemical variation on the evolution of the Néel temperature and NV signals.

Answer:

The gradual color change across the specimen in Supplementary Fig. 3d is a thickness effect due to TEM specimen preparation. In particular, classical TEM lamella preparation, which we applied for the samples discussed in this work, leads to wedge-shaped TEM lamella, not plane-parallel ones. This means that the TEM lamella thickness is gradually changing. As a consequence, for cross-sectional specimens, the lamella thickness gradually increases from the sample surface (in our case the Pt layer) into the direction of the substrate, but also in the lateral sample direction. More specifically, for the element map shown in Fig. S3d, the lamella thickness increases from the upper right side of the image to the lower left side. However, an increasing lamella thickness results in enhanced EDX signal intensity, thus leading to a gradual color change. For the thinner chromium oxide sample, there is also such gradual lamella thickness change. However, the field of view shown in Supplementary Fig. 6d is significantly smaller than that of Supplementary Fig. 3d. Consequently, the lamella thickness change across this smaller field of view is significantly smaller, leading to a more homogeneous EDX signal intensity and hence a homogeneous color distribution for Supplementary Fig. 6d.

Regarding the Pt/ Cr_2O_3 interface: there is no Cr-enriched layer for both samples, the 30-nm and the 250-nm-thick one, within the resolution of the analysis.

According to the STEM-EDX results, we do not see any hint for an O deficiency at the Pt/ Cr_2O_3 interface. These remarks are added to the Supplementary Note 1 (page 2).

Indeed, Cr_2O_3 can exhibit ferromagnetic moment due to defects [14–17] as reported for many other oxides [18]. To address this remark, we performed SQUID-VSM measurement on the Cr_2O_3 thin film sample with the thickness of 50 nm. The Al_2O_3 substrate was magnetically characterized prior to the deposition of the Cr_2O_3 (50 nm) thin film. The measured magnetization curve is shown in Supplementary Fig. 43. The obtained value of the saturation magnetization of about 0.5 emu/cc is one order of magnitude smaller than the one indicated in the reference cited by Reviewer for the sample of similar thickness.

However, we note that even if such ferromagnetic signal would be present, we do not expect related modifications of the NV microscopy and transport measurements for the following reasons: (i) The Hall resistance measurements are sensitive to the near-surface layer [19], while the defect-induced ferromagnetism is related to the film interior. (ii) The sensitivity of NV magnetometry is dependent on the spatial scale of the spatial inhomogeneity of the respective magnetic texture. A weak anisotropy for M_s^{def} suggests domains and domain walls much larger than ones related to the antiferromagnetic ordering in Cr_2O_3 . Therefore, defect-induced ferromagnetism in our setup for NV measurements should be hidden by contribution related to Cr ions.

Comment 3b:

Authors should perform systematic in-plane and out-of-plane SQUID magnetometry to rule out the presence of spurious FM contribution in all the samples.

Answer:

To address this comment of the reviewer we have performed a careful investigation of the magnetic properties of Cr_2O_3 thin film sample using a SQUID-VSM device. Prior to the deposition of the Cr_2O_3 (50 nm) thin film, the Al_2O_3 substrate was magnetically characterised. The magnetic properties of both the Al_2O_3 substrate and the thin film sample were investigated in in-plane applied magnetic field geometry. The measurements revealed a linear dependence of the magnetic moment on the applied magnetic field, which corresponds to the diamagnetic Al_2O_3 , antiferromagnetic Cr_2O_3 and paramagnetic background coming from the quartz sample holder and GE-varnish glue used for sample mounting. To check the presence of ferromagnetic response in the sample, we fit and subtract the linear component of the signal. The data is shown in Supplementary Fig. 43. We see only drifts during the measurement but nothing similar to the ferromagnetism shown in the references mentioned by the Reviewer.

This information is added as new Supplementary Fig. 43 and in Methods A (page 5).

Supplementary Fig. 43. **Superconducting quantum interference device – vibrating sample magnetometry (SQUID-VSM) characterization of the 50-nm-thick Cr_2O_3 thin film.** (a) Magnetic hysteresis measured of the sample at different temperatures. The sample reveals linear change of magnetic moment with the applied magnetic field, which is expected for the diamagnetic Al_2O_3 substrate and antiferromagnetic (or paramagnetic above Néel temperature) Cr_2O_3 thin film. (b) Same as panel (a) but with subtracted linear contribution. The data reveal only a small variation of the magnetic moment below $0.2 \mu\text{emu}$, which we consider as a detection limit of the device.

Comment 3c:

Addressing the question by Reviewers 1 and 2, authors cite the reference *Acta Materialia* 61, 2703 (2013) and state that above a certain thickness of the layer, it is energetically favourable to induce dislocations at the top layer, which then diffuse into the interior of the film. While the first part of this sentence is agreeable, the second part is not that clear-cut. The cited reference mentions (on page 2709) that “Once the dislocations are nucleated at the free surface, their propagation to the film–substrate interface is determined by the lattice frictional stress associated with dislocation glide in the film. In relatively soft materials, the propagation of dislocations is not the rate-limiting step for the relaxation process. However, in hard materials, such as III-nitrides and II-oxides, the propagation of dislocations plays a critical role by delaying the relaxation process.” This suggests that the dislocation propagation is a strongly material-dependent and temperature-dependent property. Moreover, the cited reference does not discuss any specific structural/strain evolution of Cr_2O_3 films on Al_2O_3 at all. Hence, the claim that bottom-most layers in thicker Cr_2O_3 films (100 and 250 nm) are strain-free or pseudomorphic-layer-free still remains unsubstantiated.

Answer:

The presence of the pseudomorphic layer is evident in the RSM images as the specific streak-like feature at the lattice position of Al_2O_3 on the RSM image. This feature is present for thinner samples (30- and 50-nm-thick samples) and is absent for samples of 100 and 250 nm thickness. The presence of these features in RSM images indicate that there is a Cr_2O_3 layer, which has in-plane lattice parameter as the Al_2O_3 substrate. This layer is referred to as the pseudomorphic layer.

The mentioned references are cited to provide information to the reader on the mechanism of dislocation formation and their propagation. We agree that in the content how Reviewer read the sentence, the references could be misleading. Therefore, we deleted them in the revised version to avoid misunderstandings.

As it was suggested by the Reviewer in Comment 3f, we performed additional HAADF-STEM measurements for the 30-nm-thick sample, which clearly shows the strain gradient along the film thickness (see the respective answer below). Below, we present separate answers to these questions.

Comment 3d:

Authors have added an HR-TEM image in SI Fig 2c, but its clarity is poorer in comparison to that of main Fig 1c or SI Fig 5a. Hence, it is difficult to judge the indicated positions of the misfit dislocations and absence/presence of a pseudomorphic layer.

Answer:

Indeed, the quality of that SI figure was poor. The reason is the compression of the document to make it less than 20 MB size for submission. The original TEM image is of high quality. We do not compress the supporting information file for this submission.

Furthermore, to make this aspect clearer, we provide an inset with zoomed region containing dislocations (see Supplementary Fig. 2c).

Comment 3e-1:

Authors assert in response to Reviewer 1 that RSM characterization is the most appropriate method to unambiguously determine thickness-dependent lattice parameters. However, the references provided [JAP 125, 082508 (2019); APL 95, 242902 (2009); AEM 4, 1800106 (2018); Acta Materialia 58, 6665 (2010)] primarily seem to use RSM analysis to determine either averaged lattice parameters of that sample or lattice parameters of multiple crystal fractions with clearly distinguishable and resolvable reciprocal space peaks. However, this is not at all the case with 1D multi-peak fitting deployed by the authors to resolve z-dependent sample fractions in the non-pseudomorphic regions, in SI Figs 8,10,12.

Answer:

To address this comment of the Reviewer, we performed an entirely new series of XRD measurements of the same series of samples but using another XRD device. These new data are summarized in Supplementary Note 2 (pages 3-4) and presented as new Supplementary Figures 13,14,17,18,19,46,47.

These new data agree fully with the old XRD study already presented in the manuscript (Supplementary Figure 26). We clearly see features of the pseudomorphic layer for thinner samples (streaks are located exactly at the in-plane lattice position of Al_2O_3 and deformed reflexes from interior of the Cr_2O_3 thin films).

The mentioned references were inserted in the supporting information file to indicate that this kind of deformed reflexes are typically observed. Again, we apologize that from the content the statement could be erroneously interpreted that the authors of that papers also performed fine analysis of the peaks to assess the distribution of strain in their samples.

At the same time, the mentioned papers use the complex-shaped peaks in their RSM images to extract lattice parameters of different crystalline fractions within the sample. Technically, we do the same: we slice the complex-shaped peak in several segments and perform analysis of these segments independently. The difference with the mentioned papers is that we do not claim that in our samples we have different crystalline fractions. We expect (and TEM data supports this expectation) that our samples have the same crystalline structure yet with gradually changing lattice parameters.

Comment 3e-2:

It is evident from the figures below that the same line cut can be fitted in numerous ways (other than what is shown in SI Fig 8h), resulting in very different sample fractions and lattice constants. Moreover, surprisingly, the background of the peaks is not zero, in SI Figs 7-12, which adds further flexibility in the fitting process. In total for a fit involving double/triple peaks there are 7/10 free parameters, respectively. Also, authors do not discuss why they use 1D instead of the usual 2D Gaussian fitting of an RSM peak. Overall, despite the fact that the structural trends in SI Table 2 make general qualitative sense, authors' current arguments have *failed to convince me* that the extensive (*semi*)*quantitative* fitting deployed in SI Figs 8,10,12 is clear or unambiguous.

Answer:

We show the new set of XRD measurements (Supplementary Figures 13,14,17,18,19,46,47) where we explicitly provide the parameters of the fits for each peak.

Experimental RSM data of the asymmetric $(10\bar{1}10)$ reflections taken of thin films for the structural analysis (fitting area is indicated with the white square in all figures). The line scans q_x and q_z were used for the calculation of the lattice parameters. Using a single Gaussian function, we extracted the central position of each peak, which is taken as the average values of $q_{x,av}$ (nm^{-1}) and $q_{z,av}$ (nm^{-1}) and correspond to the in-plane lattice parameter a (nm) and out-of-plane lattice parameter c (nm), respectively.

Each of these fittings (except the thin films grown on a Pt seed layer) suggests that the single peak fit is not sufficient to describe the data. Therefore, we used multipeak Gaussian function to distinguish different zones in the sample. The q_x and q_z positions corresponding to the centers of the peaks are used as a guide to fit the line scans. The multipeak Gaussian fits to the data allow determining the lattice parameters of the different zones in the sample. For analysis of the structural inhomogeneity of the sample, we did the following:

- Each peak was divided into several slices every 0.04 nm^{-1} along the respective axes.
- We used a multipeak Gaussian function fitting of each slices with the fixed position of the respective zones obtained from the fitting discussed above.
- Based on the obtained results we assumed a structure of the film.

The background of the peaks in the new XRD data is zero as it should be. Regarding the presentation of the XRD data in the previously submitted manuscript: it was our mistake with plotting the curves with an offset. We apologies for this mistake.

If a reciprocal space map (RSM) contains only peaks from homogeneous materials, an automatic peak search and fitting (as performed by Rigaku's software "SmartLab Studio II") will return the exact peak positions see figure below (a). Supplementary Fig. 46 and 47 show RSM data of a relaxed Cr_2O_3 film on a single-crystalline Al_2O_3 substrate with a Pt buffer layer and the result of an automatic peak search with default parameters.

However, if the film is subject to an inhomogeneous strain, an adequate single peak is hard to define see figure below (b). Thus, we decided to analyze profiles along the q_x and q_z directions instead, which give a better understanding of how the intensity is distributed. A comparison between the positions of the main film peak fitted by "SmartLab Studio II" on the 2D data set and by "Origin" software on extracted profile confirms good agreement with deviations of less than 0.002 nm^{-1} . In both cases, a symmetric Gaussian function was used according to the "SmartLab Studio II" manual. Thus, we believe that the approach of analyzing individual profiles yields valid peak positions similar to the automatic processing of the 2D data set (where applicable), while providing a more detailed insight into effects like pseudomorphic growth.

Furthermore, as suggested by the Reviewer, we performed additional high-angle annular dark field scanning TEM images (HAADF-STEM) measurements, which support the conclusions on the strain distribution obtained from the analysis of the RSM data.

Automatic peak search and fitting performed by "SmartLab Studio II" (a) The RSM of the $(10\bar{1}10)$ reflection of a 250-nm-thick Cr_2O_3 thin film grown on Pt(20 nm) seed layer on Al_2O_3 substrate. The software can easily locate peaks from the strain-free thin film and from the substrate. (b) The RSM of the $(10\bar{1}10)$ reflection of a 50-nm-thick Cr_2O_3 thin film grown on Al_2O_3 substrate. The software uses symmetric Gaussian function in order to provide peak position. The insert shows automatically calculated peak position, which is consistent with our refinement.

Comment 3f:

Given the foundational nature of this work in the antiferromagnetic community, structural experiments/analyses play an important role in estimating strain evolution and piezo- vs flexo-effects. Hence, I would have expected the authors to perform careful z-dependent lattice parameter quantification across all samples by other experimental means which are *direct* and *unambiguous*. One example is aberration-corrected scanning transmission electron microscopy in the high-angle annular dark field imaging (STEM-HAADF), which can provide atomic resolution and is much superior to HR-TEM. Such an approach has been nicely deployed in previous flexo-effect studies to characterize the z-dependent strain evolution [see Science 366, 475 (2019)]. In principle, such an experiment would be straightforward to perform with the type of the samples in the present manuscript, and it would immediately put to rest all the concerns raised above in comments 3c-3e.

Answer:

To characterize the interface between Al_2O_3 and Cr_2O_3 of 30 nm thickness, we acquired aberration-corrected high-resolution high-angle annular dark field scanning TEM images (HAADF-STEM; Supplementary Fig. 5), exhibiting atomic column contrast roughly proportional to Z^2 (Rutherford cross section with Z being the atomic number). Therefore, the O column remains invisible in this acquisition mode. The data reveal an abrupt change between weakly scattering Al ($Z = 12$) and Cr ($Z = 24$), and hence an atomically sharp interface. Moreover, the data confirms epitaxial phase matching between Al_2O_3 and Cr_2O_3 with the same in-plane lattice parameter at the interface.

To quantify the strain distribution along the whole thickness of the Cr_2O_3 film, we conducted a geometric phase analysis [20][21] of large field of view HRTEM data (Supplementary Fig. 5a). The Fourier transform, shown as inset in Supplementary Fig. 5a, confirms crystallinity of the Cr_2O_3 layer. Supplementary Fig. 5b,c displays the diagonal components u_{xx} and u_{zz} , revealing the varying in- and out-of-plane dilation or compression (the shear component u_{xz} was constant throughout the image (Supplementary Fig. 5d). In particular, we observe an almost linearly increasing in-plane strain tensor component u_{xx} as a function of the distance from the interface until bulk lattice parameter in agreement with the XRD data. This gradual relaxation of the Cr_2O_3 lattice clamped at the Al_2O_3 interface is facilitated by the presence of misfit dislocations and other crystal defects, appearing as sharp lines of diverging strain in the strain maps (indicated with circles). The out-of-plane lattice constant of Cr_2O_3 , being not clamped, shows a sharp jump at the interface, which is subsequently reduced again in response to the growing in-plane lattice parameter.

For the geometric phase analysis, we selected the two linearly independent systematic reflections $(0\bar{1}1\bar{2})$ and $(0\bar{1}14)$ for retrieving their geometric phase, and hence displacement field along the corresponding reciprocal lattice direction. Here, the mask sizes utilized for cutting out the reflection had a characteristic radius of 0.5 nm^{-1} , corresponding to 2 nm spatial resolution of the displacement field maps. Finally, the elastic strain tensor is evaluated utilizing Eq. (S10). Consequently, the ratio between the in- and out-of-plane lattice parameters of Cr_2O_3 (Supplementary Fig. 23e) and the strain tensor (S24) reads $a = (1 + u_{xx})a_0$ and $c = (1 + u_{zz})c_0$. We used the bulk Cr_2O_3 lattice parameters a_0

and c_0 as reference, with respect to which the strain is calculated. To retrieve the averaged strain profiles as function of distance to the Al_2O_3 interface, we deliberately neglected the regions of diverging strain (due to crystal defects) by considering only strain components smaller than 0.1 in the evaluation. The presence of the crystal defects in combination with the numerical derivative is responsible for the large scatter of the strain data.

We note that to be consistent with HAADF-STEM, the notation for the in-plane changed to have u_{xx} negative for compressive strain.

Supplementary Fig. 5. **High-resolution TEM analysis of the strain state in a 30-nm-thick Cr_2O_3 thin film.** (a) Large field of view high-resolution TEM image taken at $(2\bar{1}\bar{1}0)$ zone axis. Top-right inset displays the Fourier transform, including the two systematic reflections $(0\bar{1}1\bar{2})$ and $(0\bar{1}14)$ employed for geometric phase strain analysis. Bottom-right inset shows a high-resolution HAADF-STEM image taken at $(12\bar{3}0)$ crystal orientation of the interface region. Note, the distortion of the atomic column positions in the lower part of the HAADF-STEM image are due to image drift. (b-e) 2D strain maps of the region displayed in (a), representing different components of the strain tensor: (b,c) diagonal components u_{xx} and u_{zz} , (d) shear component u_{xz} and (e) rotation component w_{xz} . Lines of diverging strain in the strain maps are indicated with circles.

Supplementary Fig. 23. **HR-TEM strain analysis of the strain gradient on a 30-nm-thick Cr_2O_3 thin film.** Diagonal components of the strain tensor (a) u_{xx} and (b) u_{zz} reveal strain gradient from the bottom to the top film interface. (c) Shear component u_{xz} and (d) rotation component w_{xz} are almost constant along the film thickness. (e) Averaged profile of the ratio between in-plane and out-of-plane lattice constants (c/a) obtained from the analysis of HAADF-STEM image shown in Fig. 5a

Comment 4:
Modelling

Comment 4a:

Authors have clarified in response to Reviewer 1 that the piezoeffect contribution is determined by the average strain. However, it is experimentally determined from the shift of the magnetotransport-derived T_N^{top} relative to T_0 (see SI Note 7, SI Fig 32, SI Table 3). This does not seem correct as the strain experienced by the top surface is not the average strain, but the lowest strain (authors state this themselves in main Fig 4 caption). Based on the definition adopted by the authors, I would have expected the piezo-effect to be determined by $(T_N^{\text{top}} + T_N^{\text{bot}})/2$, for samples hosting linear decrease of T_N with thickness. Why is this not the case? Authors need to adopt a consistent framework tying theoretical and experimental analyses to avoid misinterpretation.

Answer:

We apologize for this inconsistency, which is now corrected in the revised Supplementary Information file.

Comment 4b:

SI Table 3 caption now states ΔT_{piezo} is estimated from Eqn S15 (DFT result), but the value given is from $T_N^{\text{top}} - T_0$ (experimental result). There is clearly some error here. Is there a mistake on the right most side of Eqn S14, as it seems to double count T_0 ?

Answer:

We apologize for this inconsistency, which is now corrected. The correct value of ΔT_{piezo} is calculated by DFT and presented in revised Supplementary Table 3.

Comment 4c:

After Eqn S22, authors mention that $M_{\text{fm},x}$ is averaged to zero across the film area. Does this mean that the sign and/or magnitude of this contribution would vary laterally across the sample? In principle, such variations could result in NV contrast if they are not too gradual. Hence, authors should discuss why these are not observed.

Answer:

The discussion about M_{fm} is updated according to the question of the Reviewer 1 (Supplementary Information; page 15). We also restored the missed term in the expression for $M_{\text{fm},x}$, which suggests absence of in-plane components of M_{fm} if the strain gradient is applied along the c axis:

$$\begin{aligned} M_{\text{fm},x} &= -\mu_{xxxz} \left(\frac{\partial u_{xx}}{\partial z} - \frac{\partial u_{yy}}{\partial z} \right) - 2\mu_{xzzz} \frac{\partial u_{xz}}{\partial z}, \\ M_{\text{fm},y} &= 2\mu_{xxxz} \frac{\partial u_{xy}}{\partial z} - 2\mu_{xzzz} \frac{\partial u_{yz}}{\partial z}, \\ M_{\text{fm},z} &= -\mu_{xxxz} \left(\frac{\partial u_{xx}}{\partial z} + \frac{\partial u_{yy}}{\partial z} \right) - \mu_{zzzz} \frac{\partial u_{zz}}{\partial z}, \end{aligned} \quad (4)$$

In absence of shear ($u_{xz} \equiv u_{xy} \equiv u_{yz} \equiv 0$) and uniform in-plane strain, the magnetization is proportional to the coefficients μ_{xxxz} and μ_{zzzz} , and aligned with the c axis.

Comment 4d:

Eqn S22 and S27 state that while $M_{\text{fm},y}=0$, $M_{\text{fm},x} \neq 0$ but weak. Is this due to crystallographic constraints in bulk Cr_2O_3 ? Are Cr_2O_3 thin films expected to have sizable in-plane anisotropy to result in such crystallographic constraints?

Answer:

For the case of Cr_2O_3 and strain gradient along the c axis, the non-trivial magnetization component is expected to be aligned with c axis (z axis in the laboratory reference frame).

Comment 5:**Other Comments:****Comment 5a:**

Authors use similar sounding terms across different contexts: e.g. NV contrast, stray fields, NV signal, $\mathbf{B}_{\text{NV}} = \mathbf{B}_{\text{NVred}} - \mathbf{B}_{\text{NVblue}}$ and also $\mathbf{B}_{\text{NV}} = \langle \mathbf{B}_{\text{top}} \rangle + \langle \mathbf{B}_{\text{bot}} \rangle + \langle \mathbf{B}_{\zeta} \rangle$, in discussions/figures. Based on the discussion in the methods I surmise that they are all related, but they may imply different things. Other examples include the term μ_0 (permeability vs flexomagnetic coefficient), and the terms μ_z vs μ_{zz} . Authors should readjust the nomenclature, where possible, to avoid confusion.

Answer:

We thank the Reviewer for this remark. We adjusted the nomenclature for better clarity. Namely, the term “NV contrast” is used for the difference between the average value of stray fields in oppositely magnetized domains and marked as \tilde{B} (page 8, main text), “NV signal” is used for B_{NV} . The term μ_0 is kept for the magnetic permeability and the estimation of the flexomagnetic coefficient in Cr_2O_3 is labeled as μ_0^g . The on-site momentum μ_z is renamed to l_z .

Comment 5b:

In the paragraph after Eqn S6, it is stated that μ_0 value has error of $\pm 4\%$, this is less than half the value given in Eqn S6. Definitions of the terms S , δ are not provided above Eqn S6.

Answer:

We thank the Reviewer for his/her vigilant consideration. By doing due diligence on this remark, we have realized that it is more than just a typo but an unfortunate mistake on our side. This error has been fixed in the revised version of the manuscript. We also introduced a better consistency in definitions between SI Note 6 and SI Note 7. The proper reading of the Eq. S6 (Eq. (S9) after revision) is

$$\mu_0^g = \frac{\Delta l_z}{S(\Delta u_{xx} + \Delta u_{zz})} = (12.0 \pm 0.5) \mu_{\text{B}}/\text{nm}^2,$$

where S is unstrained Cr_2O_3 unit cell surface area in the ab -plane. In this respect, Supplementary Tables 4 and 5 are updated.

Comment 5c:

In SI Fig 8 the line cuts shown in panel c and d, do not seem to correspond clearly to the RSM map in panel b. For example, line cut F in panel c has a sharp peak near ~ 2.4 /nm, but this is not clearly seen along line cut F in panel b. Similar issues are also present in SI Fig S10. Is this an error in the captioning?

Answer:

We thank the Reviewer for noting this inconsistency. The figures are carefully revised.

Comment 5d:

Is there a typo in confidence intervals of the c and a lattice parameter values in SI Table 2? Also, c/a ratio error doesn't seem to follow standard error propagation. Do authors use a non-standard approach? Authors should include the data for 250 nm sample and bulk data in the table as well.

Answer:

The value for the c/a error is 0.025. Having the confidence intervals δa and δc for a and c values, respectively, the confidence interval for c/a was calculated as

$$\delta \frac{c}{a} = \frac{c\delta a + a\delta c}{a^2}. \quad (5)$$

Comment 5e:

The caption seems to be missing for SI Fig 40c?

Answer:

We thank the Reviewer for pointing this out. The caption is added.

Comment 5f:

There are multiple occasions across the manuscript/SI/Figures, where the source of the error bars has not been explained.

Answer:

Following this remark of the Reviewer, we provide further information on the error bars. In particular:

1. In Fig. 1d,e and in Supplementary Figs. 20–21, error bars correspond to the standard deviation of the fit. In Supplementary Figs. 20–21 some error bars are smaller than symbols. The corresponding comment is added in Supplementary Note 3, page 6.
2. The error bars on the estimated in-plane strain in Figure 1f and in Supplementary Fig. 25 indicate the uncertainty of the peak fit of the Raman spectra. The corresponding comment is added in the Supplementary Note 4, page 7.
3. Fig 1f, Fig 1ij and Supplementary Fig. 26: The error bars of the lattice parameters estimated by XRD are calculated based on the instrumental precision of the diffractometer and from uncertainty of the peak fitting. The corresponding comment is added in the Supplementary Note 2, page 4.
4. Fig 2d: error bars on the T_N^{top} assessed by magnetotransport measurements arise from the smooth decay of the $|R_0(T)|$ to the noise level in the vicinity of the AFM-PM transition. The corresponding comment is added in the Supplementary Note 5, page 8.
5. Fig. 3b-d: for the NV study, the detection limit is $5 \mu\text{T}$ and the error of NV contrast is about $2 \mu\text{T}$ (corresponding to the halfwidth of the distribution). The corresponding comment is added in Methods, page 8.

-
- [1] E. A. Eliseev, M. D. Glinchuk, V. Khist, V. V. Skorokhod, R. Blinc, and A. N. Morozovska, Linear magnetoelectric coupling and ferroelectricity induced by the flexomagnetic effect in ferroics, *Physical Review B* **84**, 174112 (2011).
 - [2] B. A. Belyaev, A. V. Izotov, P. N. Solovev, and N. M. Boev, Strain-gradient-induced unidirectional magnetic anisotropy in nanocrystalline thin permalloy films, *physica status solidi (RRL) – Rapid Research Letters* **14**, 1900467 (2020).
 - [3] I. Veremchuk, P. Makushko, N. Hedrich, Y. Zabala, T. Kosub, M. O. Liedke, M. Butterling, A. G. Attallah, A. Wagner, U. Burkhardt, O. V. Pylypovskiy, R. Hübner, J. Fassbender, P. Maletinsky, and D. Makarov, Magnetism and magnetoelectricity of textured polycrystalline bulk Cr_2O_3 sintered in conditions far out of equilibrium, *ACS Applied Electronic Materials* [10.1021/acsaelm.2c00398](https://doi.org/10.1021/acsaelm.2c00398) (2022).
 - [4] I. Veremchuk, M. O. Liedke, P. Makushko, T. Kosub, N. Hedrich, O. V. Pylypovskiy, F. Ganss, M. Butterling, R. Hübner, E. Hirschmann, A. G. Attallah, A. Wagner, K. Wagner, B. Shields, P. Maletinsky, J. Fassbender, and D. Makarov, Defect nanostructure and its impact on magnetism of $\alpha\text{-Cr}_2\text{O}_3$ thin films, *Small* **18**, 2201228 (2022).
 - [5] T. Kosub, M. Kopte, R. Hühne, P. Appel, B. Shields, P. Maletinsky, R. Hübner, M. O. Liedke, J. Fassbender, O. G. Schmidt, and D. Makarov, Purely antiferromagnetic magnetoelectric random access memory, *Nature Communications* **8**, 13985 (2017).

- [6] R. Schlitz, T. Kosub, A. Thomas, S. Fabretti, K. Nielsch, D. Makarov, and S. T. B. Goennenwein, Evolution of the spin Hall magnetoresistance in Cr₂O₃/Pt bilayers close to the Néel temperature, *Applied Physics Letters* **112**, 132401 (2018).
- [7] A. Mahmood, W. Echtenkamp, M. Street, J.-L. Wang, S. Cao, T. Komesu, P. A. Dowben, P. Buragohain, H. Lu, A. Gruverman, A. Parthasarathy, S. Rakheja, and C. Binek, Voltage controlled néel vector rotation in zero magnetic field, *Nature Communications* **12**, 10.1038/s41467-021-21872-3 (2021).
- [8] P. Appel, B. J. Shields, T. Kosub, N. Hedrich, R. Hübner, J. Faßbender, D. Makarov, and P. Maletinsky, Nanomagnetism of magnetoelectric granular thin-film antiferromagnets, *Nano Letters* **19**, 1682 (2019).
- [9] S. Foner, High-field antiferromagnetic resonance in Cr₂O₃, *Physical Review* **130**, 183 (1963).
- [10] Y. Dovzhenko, F. Casola, S. Schlotter, T. X. Zhou, F. Büttner, R. L. Walsworth, G. S. D. Beach, and A. Yacoby, Magnetostatic twists in room-temperature skyrmions explored by nitrogen-vacancy center spin texture reconstruction, *Nature Communications* (2018).
- [11] N. Hedrich, K. Wagner, O. V. Pylypovskiy, B. J. Shields, T. Kosub, D. D. Sheka, D. Makarov, and P. Maletinsky, Nanoscale mechanics of antiferromagnetic domain walls, *Nature Physics* **17**, 574 (2021).
- [12] M. S. Wörnle, P. Welter, M. Giraldo, T. Lottermoser, M. Fiebig, P. Gambardella, and C. L. Degen, Coexistence of Bloch and Néel walls in a collinear antiferromagnet, *Physical Review B* **103**, 094426 (2021).
- [13] T. van der Sar, F. Casola, R. Walsworth, and A. Yacoby, Nanometre-scale probing of spin waves using single electron spins, *Nature Communications* **6**, 7886 (2015).
- [14] M. Street, W. Echtenkamp, T. Komesu, S. Cao, P. A. Dowben, and C. Binek, Increasing the Néel temperature of magnetoelectric chromia for voltage-controlled spintronics, *Applied Physics Letters* **104**, 222402 (2014).
- [15] S. Punugupati, J. Narayan, and F. Hunte, Strain induced ferromagnetism in epitaxial Cr₂O₃ thin films integrated on Si(001), *Applied Physics Letters* **105**, 132401 (2014).
- [16] S. Punugupati, J. Narayan, and F. Hunte, Room temperature ferromagnetism in epitaxial Cr₂O₃ thin films grown on r-sapphire, *Journal of Applied Physics* **117**, 193907 (2015).
- [17] M. Vila, J. Rubio-Zuazo, I. Lucas, C. Magén, A. Prados, E. Salas-Colera, I. Arnay, and G. R. Castro, Ferromagnetic epitaxial Cr₂O₃ thin films grown on oxide substrates by pulsed laser deposition, *Applied Surface Science* **534**, 147638 (2020).
- [18] L. Botsch, P. D. Esquinazi, C. Bundesmann, and D. Spemann, Toward a systematic discovery of artificial functional magnetic materials, *Physical Review B* **104**, 014428 (2021).
- [19] H. Wang, C. Du, P. C. Hammel, and F. Yang, Spin transport in antiferromagnetic insulators mediated by magnetic correlations, *Physical Review B* **91**, 220410 (2015).
- [20] M. Hÿtch, E. Snoeck, and R. Kilaas, Quantitative measurement of displacement and strain fields from HREM micrographs, *Ultramicroscopy* **74**, 131 (1998).
- [21] GPA analysis was performed using FRWRtools plugin of Ch. Koch, https://www.physik.hu-berlin.de/en/sem/software/software_frwrtools.

Reviewers' Comments:

Reviewer #1:

Remarks to the Author:

The authors have done extensive experiments to address my previous comments and questions. I am satisfied the present quality. The other referee also raised a bundle of questions. It seems the data in the present manuscript have been extensively concerned. I recommend the publication of this work as it is.

Reviewer #2:

Remarks to the Author:

The authors have addressed majority of the concerns and significantly improved the overall clarity of the manuscript. Addition of the new results has added further weight to their discoveries. The paper can now be accepted once the following minor points are cleared up:

1. SI Fig 7 shows newly added XRD scans of each sample. Similar looking broad Laue oscillations are clearly visible in all the samples (except perhaps 250nm), lying in the neighborhood of the main Cr₂O₃ (0006) Bragg reflection. While low thickness Cr₂O₃ can result in such broad fringes, the thicker Cr₂O₃ samples (50nm and beyond) would result in very narrow Laue fringes (if at all any), as seen in panel c. Hence, these broad Laue oscillations may suggest the presence of another very thin layer, which may be slightly different from the main Cr₂O₃ film. Is this the pseudomorphic Cr₂O₃ layer, (111)-oriented Pt over-layer or some other contribution entirely? Authors should clarify the origin of these broad Laue oscillations in the figure caption to avoid any confusion.

2. As already highlighted in the previous review, a direct comparison in SI Fig 41 between magnetotransport-derived ΔT_N and DFT-based ΔT_{piezo} doesn't seem internally consistent. According to authors definition and arguments, the former reports the T_N change at the top of the sample which experiences the lowest strain, whereas the latter measures this effect presumably somewhere in the middle of the sample due to the averaged strain. Hence, showing the numerical equivalence between the two seems inconsistent and confusing. Based on authors' definition shouldn't they be combining the magnetotransport and NV experimental results, to calculate $(T_{N\text{top}} + T_{N\text{bot}})/2$, to contrast against DFT-derived ΔT_{piezo} ?

3. Authors have obtained high quality HAADF-STEM data to discern the variations of the structural parameters and strain across the sample thickness in SI Fig 23. They may want to append the relevant STEM metadata along with the existing data in the main text Fig 1f-j.

We would like to thank the Reviewers for their remarks and suggestions, all of which are addressed in the revised manuscript. All changes in the manuscript are indicated in blue.

Our itemized responses to all the Reviewer's comments are below.

Reply to the Reviewer 1

Comment:

The authors have done extensive experiments to address my previous comments and questions. I am satisfied the present quality. The other referee also raised a bundle of questions. It seems the data in the present manuscript have been extensively concerned. I recommend the publicatio of this work as it is.

Answer:

We thank the Reviewer for his/her positive evaluation of our work and recommendation for publication in Nature Communications.

Reply to the Reviewer 2

Comment:

The authors have addressed majority of the concerns and significantly improved the overall clarity of the manuscript. Addition of the new results has added further weight to their discoveries. The paper can now be accepted once the following minor points are cleared up:

Answer:

We thank the Reviewer for his/her positive feedback. Following his/her suggestions, we updated the manuscript as follows:

1. We added a comment on the evaluation of Laue oscillations in XRD data.
2. We updated SI figure 41 to include another set of datapoints, which shows the increase of the average transition temperature.
3. We added a comment on the TEM data.

Comment 1:

SI Fig 7 shows newly added XRD scans of each sample. Similar looking broad Laue oscillations are clearly visible in all the samples (except perhaps 250nm), lying in the neighborhood of the main Cr₂O₃ (0006) Bragg reflection. While low thickness Cr₂O₃ can result in such broad fringes, the thicker Cr₂O₃ samples (50nm and beyond) would result in very narrow Laue fringes (if at all any), as seen in panel c. Hence, these broad Laue oscillations may suggest the presence of another very thin layer, which may be slightly different from the main Cr₂O₃ film. Is this the pseudomorphic Cr₂O₃ layer, (111)-oriented Pt over-layer or some other contribution entirely? Authors should clarify the origin of these broad Laue oscillations in the figure caption to avoid any confusion.

Answer:

We appreciate this remark of the Reviewer.

The Pt cap layer causes pronounced Laue oscillations in most of the scans shown in Supplementary Fig. 7. If we take the first three minima to the left of the Pt (111) peak, we obtain the corresponding thickness of the Pt capping layer of about 4.1 nm for the 100-nm-thick film, about 3.7 nm for the 50-nm-thick film and about 4.8 nm for the 15-nm-thick film. These values agree with the nominal thickness of the Pt capping layer. The (0006) reflection of the pseudomorphic layer is expected at about 38.3°. However, the pseudomorphic layer cannot be identified directly in Supplementary Fig. 7 due to its much lower intensity ($|F|^2$ of Pt (111) is almost 15 times larger).

For the 50-nm-thick film we observe strong Laue oscillations of a shorter but varying period, which correspond to a thickness of approximately 60 nm. A simulation shows that these oscillations with a varying period and amplitude can result from the combination of a pseudomorphic layer and a partly relaxed layer with a strain gradient. A model with homogeneous layers does not fit these oscillations.

This discussion is added to the revised manuscript (Supplementary Note 2, page 4).

Comment 2:

As already highlighted in the previous review, a direct comparison in SI Fig 41 between magneto-transport-derived ΔT_N and DFT-based ΔT_{piezo} doesn't seem internally consistent. According to authors definition and arguments, the former reports the T_N change at the top of the sample which experiences the lowest strain, whereas the latter measures this effect presumably somewhere in the middle of the sample due to the averaged strain. Hence, showing the numerical equivalence between the two seems inconsistent and confusing. Based on authors' definition shouldn't they be combining the magnetotransport and NV experimental results, to calculate $(T_{N\text{top}} + T_{N\text{bot}})/2$, to contrast against DFT-derived ΔT_{piezo} ?

Answer:

Following this suggestion of the Reviewer, we updated the SI figure 41 to include another set of points. These data compare the calculation according to Kota *et al.* (2013) with the average Neel temperature.

This remark is added to the revised manuscript (Supplementary Note 7 B, page 13) and the Supplementary Figure 41 is updated accordingly.

Comment 3:

Authors have obtained high quality HAADF-STEM data to discern the variations of the structural parameters and strain across the sample thickness in SI Fig 23. They may want to append the relevant STEM metadata along with the existing data in the main text Fig 1f-j.

Answer:

The HAADF-STEM subfigure in Supplementary Figure 5 contains high-resolution STEM data of the interface region. The strain analysis, however, was done based on the high-resolution TEM data, whose acquisition is described in the Method section.

To add the metadata of the high-resolution STEM acquisition, we revised the Supplementary Note 1 (page 2) as follows:

Instead of the sentence “**Therefore, the O column remains invisible in this acquisition mode.**” we write “**High-resolution HAADF-STEM data has been acquired at 300 kV acceleration voltage at a FEI Titan³ 80-300 TEM equipped with probe corrector. The camera length was chosen such that electrons scattered into angles between 60 mrad and 280 mrad have been detected with the annular dark-field detector. Weakly scattering O columns remain invisible at these large scattering angles.**”

Reviewers' Comments:

Reviewer #2:

Remarks to the Author:

Authors have addressed various concerns and the manuscript can now be accepted.